# AsymRnR: Video Diffusion Transformers Acceleration with Asymmetric Reduction and Restoration

Wenhao Sun [1]   Rong-Cheng Tu [§ 1]   Jingyi Liao [1 2]   Zhao Jin [1]   Dacheng Tao [§ 1]

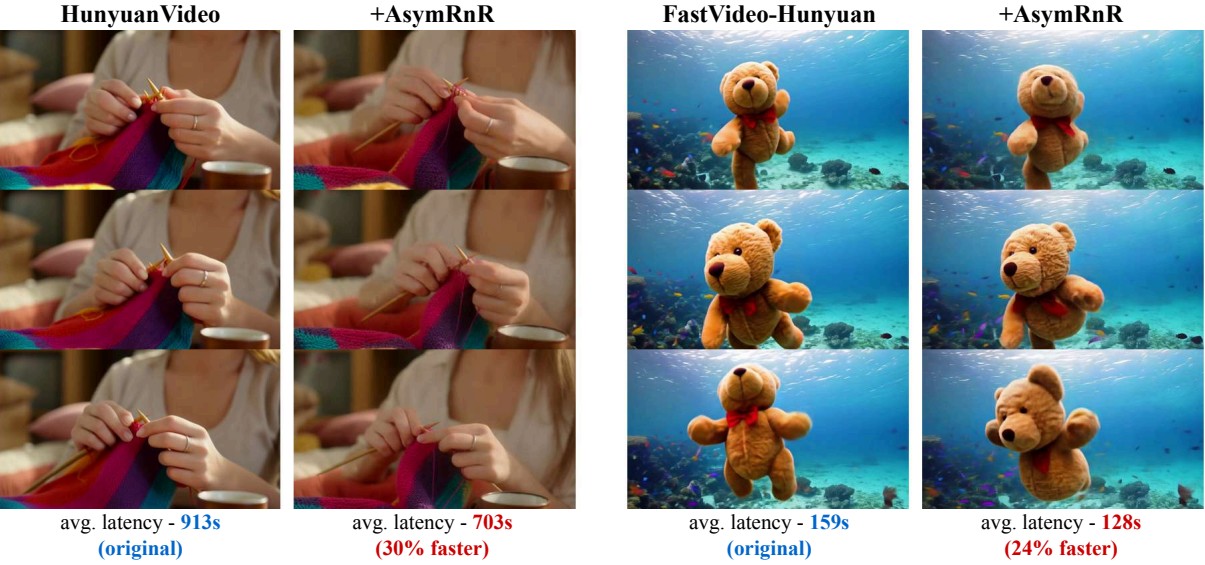

avg. latency - **913s**
(original)

avg. latency - **703s**
(30% faster)

avg. latency - **159s**
(original)

avg. latency - **128s**
(24% faster)

**prompt**: *A person is knitting ...*

**prompt**: *A teddy bear is swimming in the ocean ...*

*Figure 1.* **Quality and speed comparison between baseline models, HunyuanVideo (Team, 2024c) and FastVideo-Hunyuan (Team, 2024a), with our AsymRnR**. Our approach enables training-free, lossless acceleration for state-of-the-art video diffusion transformers.

## Abstract

Diffusion Transformers (DiTs) have proven effective in generating high-quality videos but are hindered by high computational costs. Existing video DiT sampling acceleration methods often rely on costly fine-tuning or exhibit limited generalization capabilities. We propose Asymmetric Reduction and Restoration (AsymRnR), a training-free and model-agnostic method to accelerate video DiTs. It builds on the observation that redundancies of feature tokens in DiTs vary significantly across different model blocks, denoising steps, and feature types. Our AsymRnR asymmetrically reduces redundant tokens in the attention operation, achieving acceleration with negligible degradation in output quality and, in some cases, even improving it. We also tailored a reduction schedule to distribute the reduction across components adaptively. To further accelerate this process, we introduce a matching cache for more efficient reduction. Backed by theoretical foundations and extensive experimental validation, AsymRnR integrates into state-of-the-art video DiTs and offers substantial speedup. The code is available at https://github.com/wenhao728/AsymRnR.

[1]College of Computing and Data Science, Nanyang Technological University, Singapore, Singapore [2]Institute for Infocomm Research (I2R), A*STAR, Singapore, Singapore. Correspondence to: Rong-Cheng Tu <rongcheng.tu@ntu.edu.sg>, Dacheng Tao <dacheng.tao@ntu.edu.sg>.

*Proceedings of the 42nd International Conference on Machine Learning*, Vancouver, Canada. PMLR 267, 2025. Copyright 2025 by the author(s).

## 1. Introduction

Recent progress in video generation has been largely propelled by innovations in diffusion models (Sohl-Dickstein et al., 2015; Song & Ermon, 2019; Ho et al., 2020). Building on these developments, the Diffusion Transformers (DiTs) (Peebles & Xie, 2023) have achieved state-of-the-art results across a range of generative tasks (Zhang et al., 2023; Xing et al., 2025; Shuai et al., 2024; Sun et al., 2024; Tu et al.,

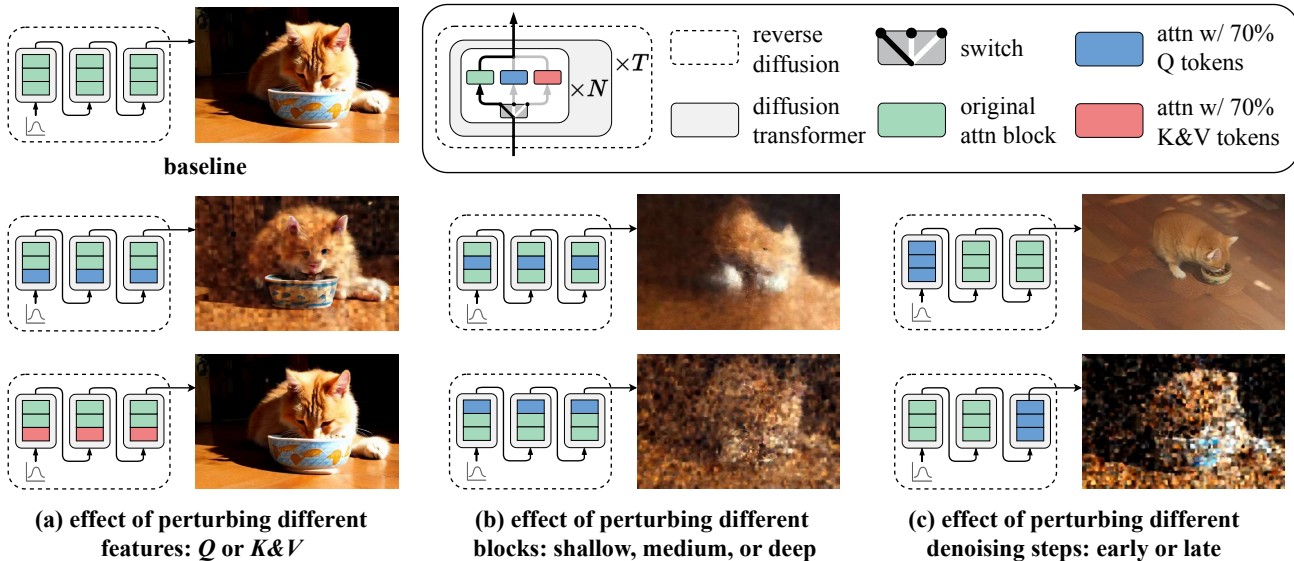

**(a) effect of perturbing different features: *Q* or *K&V***

**(b) effect of perturbing different blocks: shallow, medium, or deep**

**(c) effect of perturbing different denoising steps: early or late**

*Figure 2.* **Altering different components in video DiTs leads to varying degradation.** Green blocks represent original attention blocks. Blue blocks represent attention blocks where 30% of the query tokens are randomly discarded, allowing only the remaining 70% to contribute to the output. Red blocks represent the same perturbation applied to key and value tokens. The comparison includes perturbing: (a) different features: $Q$ or $K\&V$; (b) different DiT blocks: shallow, medium, or deep; (c) different timesteps: early or later.

2024). Despite the advancements in video DiTs, the latency remains a critical bottleneck, often taking minutes or even hours to process a few seconds of video (Lin et al., 2024; Yang et al., 2024; Team, 2024b;c).

Optimizing the efficiency of vision diffusion models has been a long-standing research focus. Distillation methods (Salimans & Ho, 2022; Meng et al., 2023; Sauer et al., 2024; Luo et al., 2023; Team, 2024c) are widely used to reduce sampling steps and network complexity. However, they require extensive training and impose high computational costs. Feature caching techniques (Zhang et al., 2024; Zhao et al., 2024; Kahatapitiya et al., 2024b) provide an alternative acceleration strategy by avoiding redundant computations in specific layers. While promising, these methods are often tailored to specific network architectures (Rombach et al., 2022; Ma et al., 2024b; Zheng et al., 2024; Lin et al., 2024), limiting their generalizability to diverse scenarios and broader model families. Another viable strategy is reducing the attention sequence length to mitigate the computational overhead in resource-intensive attention layers. For example, Token Merging (ToME) approaches (Bolya & Hoffman, 2023; Li et al., 2024) merge highly similar (*i.e.* redundant) tokens to accelerate image and video generation in Stable Diffusion (SD) (Rombach et al., 2022).

However, directly extending ToMe methods to video DiTs often results in distortions and pixelation (as shown in Figure 6). We attribute this issue to the neglect of the varying contributions of different components to the final output, and to validate this hypothesis, we randomly discard 30% tokens from different features, blocks, and denoising timesteps. The results in Figure 2 highlight three key insights: 1) Perturbations in the query ($Q$) of the early blocks significantly degrade the quality of the generation, whereas similar perturbations in the key ($K$) and value ($V$) have a less significant impact; 2) The intensity of degradation varies between the perturbed blocks; 3) Perturbing the $Q$ across all blocks but at specific denoising timesteps show that early-timestep perturbations primarily affect semantic accuracy (*e.g.* temporal motion and spatial layout), while later-timestep perturbations degrade visual details. Previous token reduction methods (Bolya & Hoffman, 2023; Li et al., 2024; Kahatapitiya et al., 2024a) apply uniform reductions across all components without accounting for their varying sensitivities. This uniformity disproportionately impacts the most vulnerable components, where even small perturbations can significantly degrade the quality of the generation. This phenomenon mirrors Liebig's law of the minimum, where ***a system's capacity is constrained by its weakest element, akin to a barrel limited by its shortest stave.***

Inspired by these observations, we propose Asymmetric Reduction and Restoration (AsymRnR) as a plug-and-play approach to accelerate video DiTs. The core idea is to reduce the attention sequence length asymmetrically before self-attention and restore it afterward for subsequent operations. We also propose a reduction scheduling that adaptively adjusts the reduction rate to account for nonuniform redundancy. Finally, we introduce the matching cache, which bypasses unnecessary matching computations to accelerate further. We conducted extensive experiments to evalu-

ate its effectiveness and design choices using state-of-the-art video DiTs, including CogVideoX (Yang et al., 2024), Mochi-1 (Team, 2024b), HunyuanVideo (Team, 2024c), and FastVideo (Team, 2024a). With AsymRnR, these models demonstrate significant acceleration with negligible degradation in video quality and, in some cases, even improve performance as evaluated on VBench (Huang et al., 2024).

## 2. Related Work

### 2.1. Video Diffusion Networks

State-of-the-art video diffusion methods (Team, 2024b;d; Yang et al., 2024; Team, 2024c) employ DiT backbones. The core module, self-attention, is defined as follows:

$$
\begin{aligned}
\mathrm{Attn}(H) &= \mathrm{softmax}\left(QK^\top\right)V \\
&= \mathrm{softmax}\left((HW_Q)(HW_K)^\top\right)(HW_V),
\end{aligned}
\tag{1}
$$

where $W_Q, W_K, W_V \in \mathbb{R}^{d \times d}$ denote the projection matrices. $H \in \mathbb{R}^{n \times d}$ represents the input sequence, $n$ represents the sequence length, and $d$ represents the hidden dimensions. Certain operations, such as scaling, positional embeddings (Su et al., 2024), and normalization (Ba et al., 2016), are omitted here for brevity.

The self-attention operation has a $O(n^2)$ time complexity. For video sequences, $n$ is typically very large. Our goal is to simplify its calculation to improve efficiency.

### 2.2. Efficient Diffusion Models

**Step Distillation.** Diffusion step distillation studies reduce sampling steps to as few as 4–8 (Salimans & Ho, 2022; Meng et al., 2023; Sauer et al., 2024). InstaFlow (Liu et al., 2024) introduces the integration of Rectify Flow (Liu et al., 2023) into distillation pipelines, enabling extreme model compression without sacrificing much quality. Consistency Models (CMs) (Song et al., 2023) propose regularizing the ODE trajectories during distillation. LCM-LoRA (Luo et al., 2023) introduces an efficient low-rank adaptation (Hu et al., 2022), facilitating the conversion of SD to 4-step models.

**Feature Cache.** Recognizing the small variation in high-level features across adjacent denoising steps, studies (Ma et al., 2024a; Wimbauer et al., 2024; Habibian et al., 2024; Chen et al., 2024) reuse these features while updating the low-level ones. T-GATE (Zhang et al., 2024) caches cross-attention features during the fidelity-improving stage. PAB (Zhao et al., 2024) caches both self-attention and cross-attention features across different broadcast ranges. Ada-Cache (Kahatapitiya et al., 2024b) adaptively caches features based on the computational needs of varying contexts.

Despite significant progress, distillation methods necessitate fine-tuning to integrate new sampling paradigms. Feature cache methods are tightly coupled with specific architectures and have not been effectively integrated into few-step sampling models. In contrast, our sequence-level acceleration method is training-free and compatible with SOTA video DiTs. Our method can also accelerate distilled models or pipelines with caching, achieving additional speedup.

### 2.3. Token Reduction

Recent advances in processing long contexts using Transformers span areas such as NLP (Leviathan et al., 2024; Xiao et al., 2024; Wang et al., 2020; Choromanski et al., 2021), computer vision (Koner et al., 2024; Rao et al., 2021; Yin et al., 2022; Choudhury et al., 2024), and multimodal tasks (Darcet et al., 2024; Li et al., 2024; Tu et al., 2023; Ma et al., 2024c; Ji et al., 2023). Many studies have focused on shortening the sequence, primarily targeting discriminative and autoregressive generation tasks. These approaches selectively truncate outputs from preceding layers, with the reductions compounding throughout the process. However, they are unsuitable for diffusion denoising, where the entire sequence must remain restorable.

**Token Merging (ToMe)** (Bolya et al., 2023) achieves token reduction by merging tokens based on intra-sequence similarity, which has been generalized to image generation with SD (Bolya & Hoffman, 2023). However, directly extending ToMe to video DiTs poses challenges, often leading to excessive pixelation and blurriness. Moreover, some designs are empirical and lack theoretical justification. We revamp their design choices from practical and theoretical perspectives for greater acceleration and consistent quality.

## 3. Method

Consider a self-attention layer that processes an input matrix $H \in \mathbb{R}^{n \times d}$, where $n$ is the sequence length and $d$ is the feature dimension. The standard scaled dot-product self-attention as shown in Equation (1) results in a $O(n^2)$ complexity, which is costly for long sequences in video generation. We accelerate self-attention in DiTs by reducing the number of tokens $n$ involved in the computation.

### 3.1. Matching-Based Reduction

The hidden states of vision transformers often exhibit redundancy (Bolya et al., 2023; Darcet et al., 2024). These observations motivate reducing the token sequence $\{h_i\}_{i=1}^n$ to a compact subsequence $\{h'_i\}_{i=1}^m \subset \{h_i\}_{i=1}^n$ prior to computation, thereby reducing computational costs by shortening the sequences. A feasible reducing strategy involves computing the pairwise similarity $[s_{ij}]_{n \times n} = [\mathrm{similarity}(h_i, h_j)]_{n \times n}$. The token pairs with the highest similarity are iteratively matched and merged, retaining a single representative token

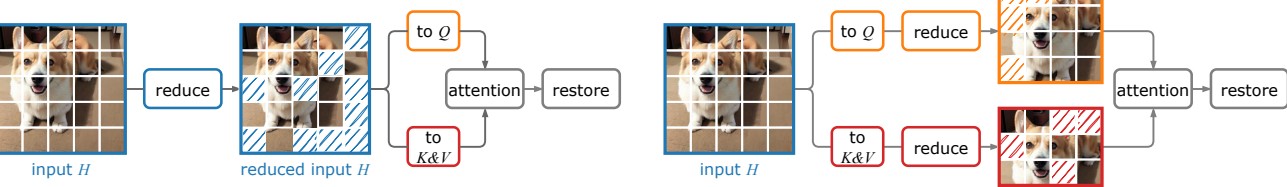

**(a) Symmetric Reduction and Restoration (SymRnR)**      **(b) Asymmetric Reduction and Restoration (AsymRnR)**

*Figure 3.* **Overview of (a) symmetric and (b) asymmetric strategies.** Both methods reduce the processing sequence length before self-attention to enhance efficiency and subsequently restore it to the original length for dense prediction. SymRnR performs reduction before mapping to $Q$, $K$, and $V$, whereas AsymRnR applies reduction afterward. This flexibility allows for the adaptive assignment of varying reduction rates to individual features. Moreover, AsymRnR supports operations on $Q$, $K$, and $V$ before reducing sequence, such as 3D rotary position embedding (Su et al., 2024), offering better compatibility. We use image patches for illustrative purposes.

per pair until the sequence is reduced to $m$ tokens.

While these methods align with intuition and perform well in practice, they lack theoretical analysis to explain their effectiveness, limiting opportunities for further optimization. From a distributional perspective, we provide a theoretical explanation motivating additional improvement. To prevent the pretrained network from being affected by covariate shift, the distribution of the reduced sequence should closely resemble that of the original sequence. Specifically, the reduction process should minimize the Kullback-Leibler (KL) divergence $D_{KL}(\mathcal{P}'||\mathcal{P})$ between the reduced sequence $\mathcal{P}'$ and the original distribution $\mathcal{P}$. Given that the analytical forms of these distributions are typically inaccessible, we employ numerical estimation techniques. Inspired by (Wang et al., 2009), we present the following corollary for estimating the KL divergence, with further details provided in the Appendix A.

**Corollary 3.1.** *Suppose $\{X_i\}_{i=1}^l$ and $\{X_i'\}_{i=1}^{l'}$ are covariance stationary sequences sampled from $\mathcal{P}$ and $\mathcal{P}'$, respectively. A Monte Carlo estimator is given by:*

$$\hat{D}_{l',l}(\mathcal{P}'||\mathcal{P}) = \frac{d}{l'} \sum_{i=1}^{l'} \log \frac{\nu(i)}{\rho(i)} + \log \frac{l}{l'-1}, \quad (2)$$

*where $\rho(i)$ is the nearest-neighbor (NN) Euclidean distance[1] of $X_i'$ among $\{X_j'\}_{j\neq i}$ and $\nu(i)$ is the NN Euclidean distance of $X_i'$ among $\{X_j\}_{j=1}^l$. The bias and variance of this estimator $\hat{D}_{l',l}(\mathcal{P}'||\mathcal{P})$ vanish as $l, l' \to \infty$.*

This theoretical framework illuminates the effectiveness of matching-based reduction methods through two key mechanisms: 1) the elimination of redundant tokens increases the dispersion of $\mathcal{P}'$, resulting in larger $\rho(i)$ values; and 2) careful control of the reduction ratio prevents excessive sparsification, maintaining small $\nu(i)$ values. This analysis not only provides theoretical validation for existing token

reduction strategies but also suggests promising directions for enhancement, which we explore in subsequent sections.

### 3.2. Prior Reduction Methods for Diffusion Acceleration

Prior works (Bolya & Hoffman, 2023; Li et al., 2024; Kahatapitiya et al., 2024a) reduce the input sequence $H$ (so symmetrically reduce $Q$, $K$, and $V$), enabling self-attention to process a shorter sequence, as depicted in Figure 3 (a). To maintain compatibility with diffusion denoising, the reduced sequences are restored to their original length by replicating each reduced token according to its most similar match among the unreduced tokens. Formally, the attention operation with Symmetric Reduction and Restoration (SymRnR) applied can be formulated as follows:

$$\text{SymRnR}(H) = (\mathcal{R}^{-1} \circ \text{Attn} \circ \mathcal{R})(H), \quad (3)$$

where $\mathcal{R}(\cdot)$ and $\mathcal{R}^{-1}(\cdot)$ represent the reduction and restoration operations, respectively. The symbol $\circ$ denotes composition operator, meaning the connected operators are applied sequentially from right to left. Notably, this process is lossy, and the resulting error is often substantial in video DiTs.

**Bipartite Soft Matching (BSM).** Computing the naive $n \times n$ similarity matrix is computationally expensive and may negate acceleration. BSM is introduced to improve the matching efficiency (Bolya et al., 2023): The tokens $\{h_1, \ldots, h_n\}$ are first partitioned into a set of source tokens $\{h_{s_1}, \ldots, h_{s_{n_1}}\}$ of size $n_1$ and a set of destination tokens $\{h_{d_1}, \ldots, h_{d_{n_2}}\}$ of size $n_2$, where $n = n_1 + n_2$. Each source token is matched with its closest destination token. Then, the top $n - m$ matched source tokens are reduced.

**Partitioning.** ToMe (Bolya & Hoffman, 2023) partitions the image tokens into chunks using 2D stride (*e.g.* $2 \times 2$) and randomly selects one token from each chunk to populate the set of destinations. We extend this approach to 3D stride (*e.g.* $2 \times 2 \times 2$) to accommodate video data.

---

[1]This generally holds for $L_p$-distances, where $1 \leq p \leq \infty$.

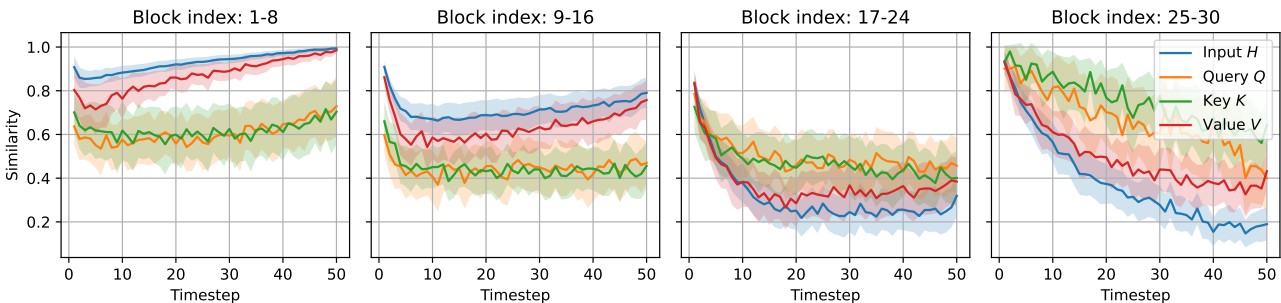

*Figure 4.* **CogVideoX ([Yang et al., 2024](#)) attention feature similarity distribution.** The shaded areas indicate the confidence interval. Blocks are divided into four groups, each exhibiting distinct trends, with variations observed across different feature types. These patterns remain consistent across generations with diverse contents.

### 3.3. Asymmetric Reduction and Restoration

Unlike previous works that focus on reducing $H$, we reduce $Q$ and $K\&V$ independently, as depicted in Figure 3 (b). In this way, the asymmetric treatment of $Q$ and $K\&V$ enables each feature to minimize its covariate shift independently during the matching process, as discussed in Section 3.1. The attention operation with our Asymmetric Reduction and Restoration (AsymRnR) is formulated as:

$$
\begin{aligned}
Q' &= \mathcal{R}_Q(Q), \quad K', V' = \mathcal{R}_{KV}(K, V), \\
\text{AsymRnR}(H) &= \mathcal{R}_Q^{-1}(\text{softmax}(Q'(K')^T)V').
\end{aligned}
\tag{4}
$$

It is worth noticing that $K$ and $V$ must share the same reduction scheme due to their one-to-one correspondence, while $Q$ can be reduced independently of the other features. Furthermore, since $Q$ acts as the "questioner" and its sequence length must match the original sequence for subsequent layer processing, while the information from $K$ and $V$ has already been encoded into the output features by attention weights, it is sufficient to restore the $Q$ sequence.

Symmetric reduction at $H$ may not necessarily achieve optimal divergence at the levels of $Q$, $K$, and $V$ but seeks to balance them. In contrast, our decoupled design allows for an asymmetrical reduction strategy and varying reduction rates across features, resulting in improved divergence for each feature and greater flexibility. Another benefit of the decoupled design is improved compatibility. For instance, techniques such as 3D rotary position embedding (ROPE) ([Su et al., 2024](#)) on $Q$ and $K$ require specific sequence lengths. SymRnR, which performs reduction before mapping to $Q$, $K$, and $V$, cannot support arbitrary reduction rates. In contrast, AsymRnR applies reduction after these operations, enabling arbitrary reduction rates and ensuring better compatibility.

### 3.4. Reduction Scheduling

Besides the asymmetric redundancy mentioned in Section 3.3, Figure 2 also reveals that perturbations in $Q$ across the shallow, middle, and deep blocks lead to varying de-

grees of degradation. Similarly, perturbations at different denoising timesteps exhibit obvious variations. This raises a natural question: how does redundancy evolve across blocks and denoising timesteps?

We examined the similarity across blocks and timesteps in video DiTs. Figure 4 illustrates temporal trends in similarity across blocks: 1) During the initial timesteps when the input to the network closely resembles random noise, higher similarity is observed across different blocks; 2) As the generation progresses, the similarity generally decreases and stabilizes; 3) The temporal trends vary significantly between blocks. For instance, in the shallow blocks, the similarity of $V$ increases steadily after the first ten steps. Whereas in other blocks, it remains relatively constant. 4) Such a pattern is model-specific but context-agnostic and can be considered an intrinsic property of the models.

To optimize computational budgets, we can reduce computations for high-similarity blocks and timesteps while maintaining low-similarity components. A key challenge is that the similarity values for the entire process are unknown in advance, complicating the decision on which components to reduce. Fortunately, we can leverage the property that similarity patterns are context-agnostic, enabling the similarity distribution to be estimated in advance through arbitrary diffusion sampling. Specifically, let $\hat{\mathcal{S}}(A, t, b)$ represent the similarity for feature $A \in \{H, Q, K, V\}$, reverse diffusion timestep $t$, and block $b$ during the advanced sampling process. Using this similarity measure, reductions can be applied selectively by thresholding $\hat{\mathcal{S}}(A, t, b)$, enabling the derivation of a scheduled reduction strategy:

$$
\tilde{\mathcal{R}}_A = \begin{cases} \mathcal{R}_A & \text{if } \hat{\mathcal{S}}(A, t, b) \geq \tau_A \\ \text{id} & \text{otherwise} \end{cases},
\tag{5}
$$

where $\tau_A$'s are the thresholding hyperparameters, which can be adjusted to optimize the trade-off between computational efficiency and output quality. $\mathcal{R}_A$ denotes the reduction operation defined in Equation (4), while id represents the identity operation (*i.e.*, no reduction). The restoration op-

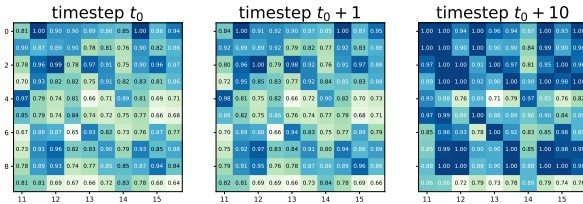

*Figure 5.* **Heatmap of matching similarity at different denoising timesteps.** The similarities across successive timesteps are nearly identical, but divergence increases with a larger step gap.

eration $\mathcal{R}_A^{-1}$ is modified correspondingly. This reduction scheduling is also asymmetric for $A \in \{H, Q, K, V\}$, allowing for precise coordination of computational resources.

Details of the hyperparameter tuning for $\tau_A$ are presented in Appendix B.

### 3.5. Matching Cache

One drawback of matching-based reduction methods is the additional cost incurred by the matching process during each reduction. Despite using BSM (Bolya et al., 2023), the matching process still significantly negates the speedup.

We observed that the matching similarity at successive denoising steps exhibits only minor differences, as illustrated in Figure 5. Similar patterns were also observed in the hidden states produced by the denoising network layers (Balaji et al., 2022; Ma et al., 2024a; Chen et al., 2024). This observation motivates us to cache the matching results over denoising steps, avoiding repeating calculations. Formally, matching similarity with caching is defined as:

$$\mathcal{S}(A, t, b) = \begin{cases} \text{BSM}(A, t, b) & \text{if } t \equiv 0 \ (\text{mod } s) \\ \mathcal{S}(A, s \cdot \lfloor t/s \rfloor, b) & \text{otherwise} \end{cases},$$

where $\mathcal{S}(A, t, b)$ denotes the matching similarity of the attention feature type $A \in \{H, Q, K, V\}$ at the reverse diffusion timestep $t$ and block $b$. $\text{BSM}(\cdot)$ represents the bipartite soft matching (Bolya et al., 2023) mentioned in Section 3.2 and $s$ denotes the caching steps. Consequently, the matching cost is proportionally reduced by $1/s$.

## 4. Experiments

### 4.1. Experimental Settings

**Models and Methods.** We exhibit experiments on Text-to-Video (T2V) task on state-of-the-art open-sourced video DiTs: CogVideoX-2B, CogVideoX-5B (Yang et al., 2024), Mochi-1 (Team, 2024b), and HunyuanVideo (Team, 2024c). We also evaluate integrating our AsymRnR approach with the step distillation approach by combining it with a 6-step distilled version of FastVideo (Team, 2024a). We focus on training-free token reduction-based acceleration and use

*Table 1.* **Quantitative comparison of CogVideoX-2B (Yang et al., 2024) with ToMe (Bolya et al., 2023) and AsymRnR.** FLOPs represent the floating-point operations required per video. Generation specifications: resolution $480 \times 720$ and 49 frames.

| METHOD | FLOPS ($\times 10^{15}$) | LATENCY (SECOND) | SPEEDUP | VBENCH ↑ | LPIPS ↓ |
|---|---|---|---|---|---|
| COGVIDEOX-2B | 12.000 | 137.30 | - | 0.8008 | - |
| + ToMe | 10.549 | 123.59 | 1.11× | 0.7825 | 0.5562 |
| + ToMe-FAST | 10.120 | 118.26 | 1.16× | 0.7732 | 0.5602 |
| + OURS | 10.342 | 121.70 | 1.13× | **0.7917** | **0.5511** |
| + OURS-FAST | **9.976** | **117.29** | **1.17×** | 0.7849 | 0.5632 |

ToMe (Bolya & Hoffman, 2023) as the baseline method. We adjust the reduction rate, align the latency, and evaluate their generation quality on CogVideoX-2B (Yang et al., 2024). Notably, ToMe is incompatible with 3D ROPE and cannot be directly integrated into the other DiTs as detailed in Section 3.3. Our AsymRnR integrates seamlessly with these methods, and we report its results across all video DiTs for comprehensive evaluation. Additional implementation details are provided in the Appendix.

**Benchmarks and Evaluation Metrics.** We follow previous work and perform sampling on over 900 text prompts from the standard VBench suite (Huang et al., 2024). It assesses the quality of generated videos across 16 dimensions. The aggregated VBench score is reported and all dimensional scores will be provided in Appendix E.2. LPIPS (Zhang et al., 2018) is used as a reference metric in the comparison analysis for semantic alignment evaluation. Note that no unique ground-truth video exists for a given text prompt; multiple generations can be equally satisfactory. Therefore, visual quality and textual alignment (measured by VBench score) are the primary performance metrics. And LPIPS is only included as a reference metric.

For efficiency evaluation, we use FLOPs and running latency[2] as metrics from both theoretical and practical perspectives. The relative speedup, $\Delta$latency/latency $+ 1$, is also provided.

### 4.2. Experimental Results

**Quantitative Comparison.** Table 1 provides qualitative comparisons between two configurations: a base version with perceptually near-lossless quality and a fast version that achieves higher speed at the cost of slight quality degradation. We set the matching cache step to $s = 5$ and the partition stride to $2 \times 2 \times 2$ for both ToMe and AsymRnR. Our higher VBench scores and lower LPIPS, achieved at comparable FLOPs and latency, demonstrate superior video quality and semantic preservation.

---

[2]Latency is measured using an NVIDIA A100 for CogVideoX variants and an NVIDIA H100 for the rest of models due to the availability of hardware at the time.

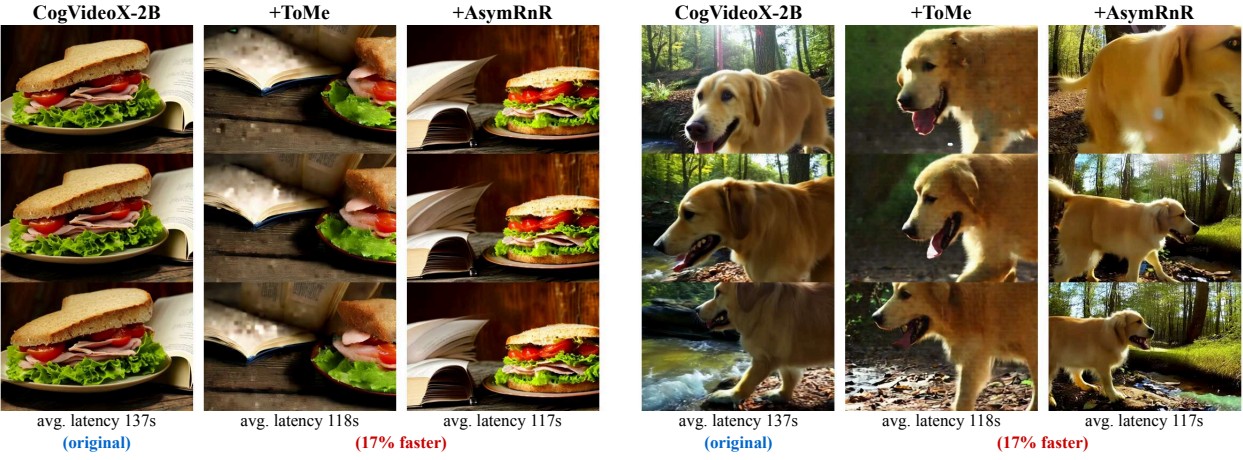

*Figure 6.* **Qualitative comparison on CogVideoX-2B (Yang et al., 2024).** ToMe (Bolya & Hoffman, 2023) exhibits blurriness (left) and pixelation (right), whereas our AsymRnR consistently performs well. The video examples are provided in the Supplementary Materials.

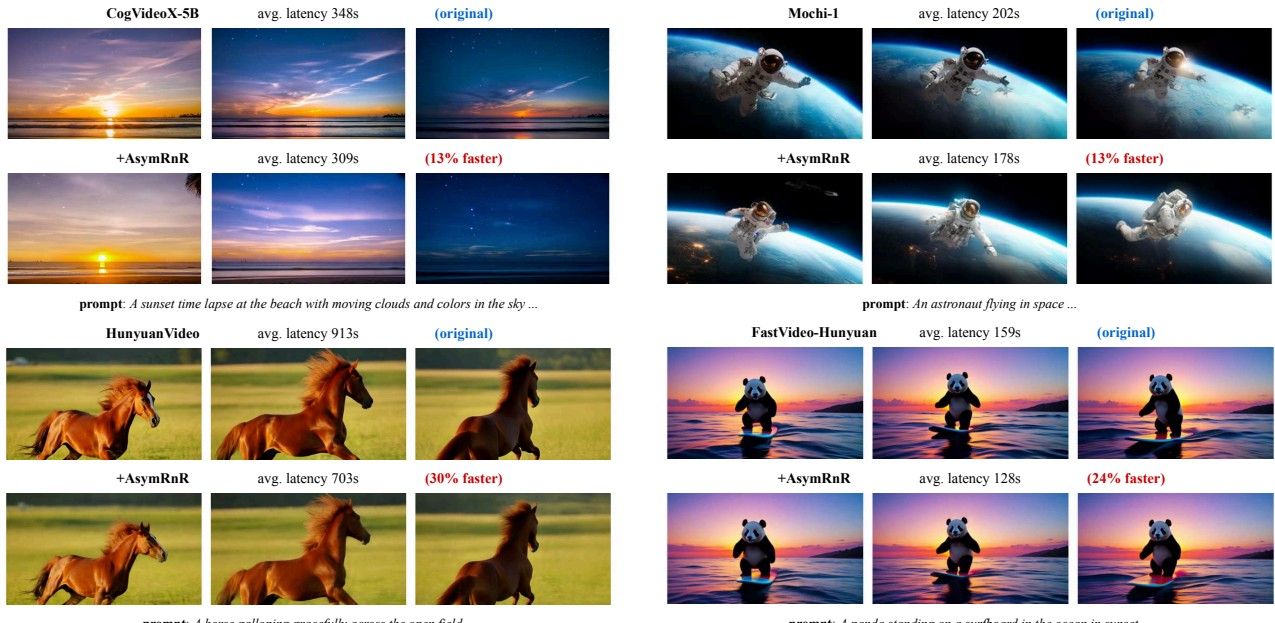

*Figure 7.* **Qualitative results on CogVideoX-5B (Yang et al., 2024), Mochi-1 (Team, 2024b), HunyuanVideo (Team, 2024c), and FastVideo-Hunyuan (Team, 2024a).** ToMe (Bolya & Hoffman, 2023) is incompatible with these models; we present videos generated by the baseline models and our proposed AsymRnR.

**Generalization to SOTA Video DiTs.** Table 2 presents additional results of applying AsymRnR to DiTs across various architectures, parameter sizes, and denoising schedulers. Applying AsymRnR to the CogVideoX-5B demonstrates less quality degradation than the 2B variant. This phenomenon is more pronounced in larger models Mochi-1 and HunyuanVideo: AsymRnR even achieves superior results over baseline models while reducing computational costs. In FastVideo-Hunyuan, a step-distilled variant of Hunyuan Video, which is capable of sampling in just 6 denoising steps, AsymRnR achieves an over 24% speedup while

maintaining perceptual quality. Our consistently strong performance underscores effectiveness and generalizability.

**Qualitative Results.** Figure 6 presents a qualitative comparison on CogVideoX-2B. ToMe generations exhibit noticeable blurriness and pixelation. Although generations with AsymRnR deviate from the baselines at the pixel level, they consistently preserve high quality and semantic coherence. Figure 7 also presents qualitative results on other video DiTs, which are incompatible with ToMe. Asym-RnR demonstrates acceleration without compromising visual quality in various baseline models and contents.

*Table 2.* **Quantitative evaluation of CogVideoX-5B (Yang et al., 2024), Mochi-1 (Team, 2024b), HunyuanVideo (Team, 2024c), and FastVideo-Hunyuan (Team, 2024a).** The generation specifications are detailed in the Appendix C.

| METHOD | FLOPs ($\times 10^{15}$) | LATENCY (SECOND) | SPEEDUP | VBENCH ↑ |
|---|---|---|---|---|
| COGVIDEOX-5B | 33.220 | 347.50 | - | **0.8074** |
| + OURS | 29.876 | 316.12 | 1.10× | 0.8061 |
| + OURS-FAST | **29.190** | **308.86** | **1.13×** | 0.8056 |
| MOCHI-1 | 35.877 | 201.85 | - | 0.7911 |
| + OURS | 32.086 | 182.64 | 1.10× | 0.7973 |
| + OURS-FAST | **31.583** | **178.26** | **1.13×** | **0.7996** |
| HUNYUANVIDEO | 128.559 | 912.98 | - | 0.8157 |
| + OURS | 104.795 | 738.18 | 1.24× | **0.8240** |
| + OURS-FAST | **100.110** | **703.29** | **1.30×** | 0.8237 |
| FASTVIDEO | 21.720 | 158.66 | - | **0.8225** |
| + OURS | 17.746 | 132.52 | 1.20× | 0.8172 |
| + OURS-FAST | **17.086** | **128.39** | **1.24×** | 0.8140 |

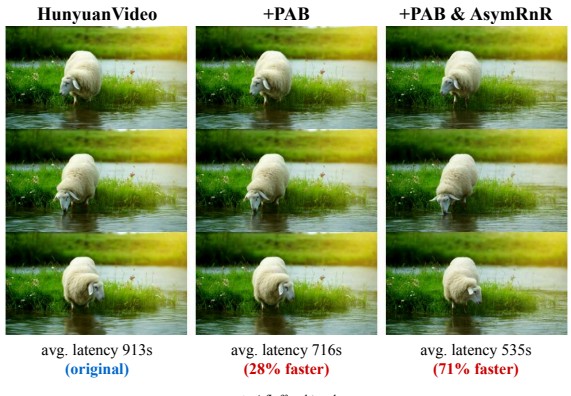

| HunyuanVideo | +PAB | +PAB & AsymRnR |
|---|---|---|
| avg. latency 913s **(original)** | avg. latency 716s **(28% faster)** | avg. latency 535s **(71% faster)** |

**prompt**: *A fluffy white sheep ...*

*Figure 8.* **Quantitative evaluation on HunyuanVideo.** AsymRnR is compatible with the feature caching method PAB (Zhao et al., 2024), and together they achieve a 1.71× overall acceleration.

**Integration with Feature Caching.** Our AsymRnR accelerates sampling by reducing the computation of the attention operator and is orthogonal to other acceleration approaches. Experiments on FastVideo-Hunyuan (Team, 2024a) demonstrate its compatibility with the step-distillation method (Wang et al., 2024). We further assess its compatibility with feature caching techniques by integrating AsymRnR with PAB (Zhao et al., 2024); the results are summarized in Table 1. When stacked on top of PAB, AsymRnR achieves a 1.71× overall speedup without compromising generation quality. The quantitative results are presented in Figure 8.

**Integration with UNet-based video diffusion models.** AsymRnR is designed to operate on attention layers, which are prevalent in diffusion models, including UNet-based (Ronneberger et al., 2015) architectures such as Animate-Diff (Guo et al., 2024). We apply AsymRnR to the spatial self-attention modules at the highest-resolution stages of

*Table 3.* **Integration with the caching-based method PAB.** AsymRnR and PAB (Zhao et al., 2024) on HunyuanVideo (Team, 2024c) further improve efficiency, achieving a total 1.71× speedup with negligible performance degradation.

| METHOD | FLOPs ($\times 10^{15}$) | LATENCY (SECOND) | SPEEDUP | VBENCH ↑ |
|---|---|---|---|---|
| HUNYUANVIDEO | 128.599 | 912.89 | - | 0.8157 |
| + PAB | 96.582 | 715.71 | 1.28× | **0.8153** |
| + PAB + OURS | **73.224** | **534.93** | **1.71×** | 0.8140 |

*Table 4.* **Integration with the UNet-based video diffusion model AnimateDiff (Guo et al., 2024).** AsymRnR achieves a 1.2× speedup. Generation specifications: 16-frame videos at $512 \times 512$ using a 50-step DDIM Euler solver (Song et al., 2021).

| METHOD | FLOPs ($\times 10^{15}$) | LATENCY (SECOND) | SPEEDUP | VBENCH ↑ |
|---|---|---|---|---|
| ANIMATEDIFF | 1.775 | 37.21 | - | 0.7748 |
| + TOME | 1.636 | 31.14 | 1.19× | 0.7693 |
| + OURS | **1.635** | **31.09** | **1.20×** | **0.7738** |

AnimateDiff. The corresponding qualitative results are provided in Table 4. This integration yields a 1.20× speedup without perceptible quality degradation.

### 4.3. Ablation Study

**Quality-Latency Trade-off for Individual Features.** Figure 9 illustrates the quality-latency curve for different feature types: $H$, $Q$, $K$, and $V$. To isolate the influence of other factors, the scheduling discussed in Section 3.4 is disabled unless explicitly mentioned otherwise in this section. We observe that reducing individual features leads to a hierarchy of $V > H > K \gg Q$. As $K$ and $V$ require identical reduction behavior, we use $V$'s matching to decide $K\&V$ reduction throughout the paper.

**Reduction Scheduling.** Table 5 presents a comparison of AsymRnR with and without scheduling. Under a capped latency budget, adaptively adjusting reducible blocks and timesteps based on the scheduling strategy significantly improves performance. Moreover, the improvement in $Q$ surpasses $V$, suggesting that $Q$ is more sensitive in low-redundancy blocks and timesteps but exhibits greater robustness to substantial reductions in high-redundancy regions. Specifically, reducing the high-similarity parts of $Q$ by 80% causes no perceptible artifacts in human evaluations. In contrast, reducing the low-similarity components by just 10% leads to noticeable distortions. It underscores the necessity of implementing such a reduction schedule.

**Matching Cache.** One major efficiency bottleneck is matching. Our newly proposed matching cache reduces the matching cost by a factor of $1/s$, but may intuitively result in

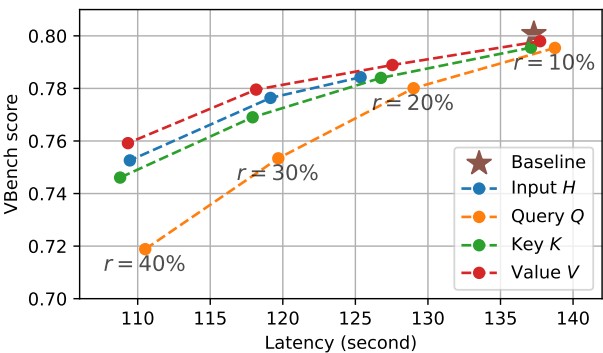

*Figure 9.* **Quality-latency trade-off for individual features.** Uniformly reducing $V$ shows superior quality, whereas reducing $Q$ in isolation leads to a substantial quality decline.

*Table 5.* **The effectiveness of the reduction schedule.** With the reduction schedule, the reductions in $Q$ and $V$ demonstrate significant improvements without increasing latency.Collaboratively reducing $Q$ and $V$ further improves performance while maintaining the same latency. Our default configurations are highlighted .

| FEATURE | SCHEDULE | FLOPs $(\times 10^{15})$ | LATENCY (SECOND) | VBENCH $\uparrow$ |
|---------|----------|----------|----------|--------|
| $Q$ | | 10.204 | 123.42 | 0.7641 |
| $V$ | | 10.276 | 122.34 | 0.7765 |
| $Q$ | $\checkmark$ | 10.620 | 123.59 | 0.7893 |
| $V$ | $\checkmark$ | 10.403 | 122.19 | 0.7787 |
| $Q+V$ | $\checkmark$ | 10.342 | 121.70 | **0.7917** |

potential quality degradation. Table 6 presents the VBench score across various caching steps. Increasing the caching step significantly reduces FLOPs and latency, with only minor quality degradation. We adopt $s = 5$ as the default configuration, balancing latency and quality.

**Similarity Metric and Reduction Operation.** Table 7 analyzes the impact of various design choices for matching and reduction. ToMe (Bolya et al., 2023) utilizes cosine similarity, which lacks the metric properties and cannot be interpreted via Corollary 3.1. Experimental results reveal that switching to (negative) Euclidean distance improves performance, offering theoretical soundness and accounting for magnitude. Additionally, ToMe's default mean-based reduction operation increases latency and causes blurriness and distortions. In contrast, our approach directly discards redundant tokens, enhancing efficiency and output quality.

## 5. Limitation

One limitation of our approach is the presence of visual discrepancies in the generated outputs as shown in Figures 6 and 7, despite maintaining semantic consistency. Additionally, acceleration and generation quality depend on hyperparameter configurations, such as the similarity threshold for

*Table 6.* **The effect of matching cache on $V$ reduction.** Increasing the caching steps does not significantly degrade performance but reduces latency when $s \leq 5$. This evaluation focuses on feature $V$ without scheduling to isolate the impact of caching steps.

| | FLOPs $(\times 10^{15})$ | LATENCY (SECOND) | VBENCH $\uparrow$ |
|---|----------|----------|--------|
| $s = 1$ | 10.210 | 134.68 | **0.7859** |
| $s = 2$ | 10.161 | 124.24 | 0.7822 |
| $s = 3$ | 9.968 | 120.77 | 0.7798 |
| $s = 4$ | 9.943 | 118.61 | 0.7763 |
| $s = 5$ | 9.917 | 118.16 | 0.7796 |
| $s = 6$ | **9.909** | **117.51** | 0.7747 |

*Table 7.* **Comparison of similarity and reduction operations.** The cosine similarity in ToMe (Bolya et al., 2023) is suboptimal compared to Euclidean similarity for AsymRnR. The mean reduction operation introduces extra processing time and degrades quality. Directly discarding yields the best results.

| SIMILARITY | REDUCTION | FLOPs $(\times 10^{15})$ | LATENCY (SECOND) | VBENCH $\uparrow$ |
|------------|-----------|----------|----------|--------|
| RANDOM | DISCARD | **9.843** | **113.79** | 0.7112 |
| DOT | DISCARD | 9.914 | 117.92 | 0.7395 |
| COSINE | DISCARD | 9.914 | 118.08 | 0.7716 |
| EUCLIDEAN | DISCARD | 9.917 | 118.16 | **0.7796** |
| EUCLIDEAN | MEAN | 9.917 | 119.90 | 0.7630 |

reduction and the reduction rate, which require tuning for each baseline model. Furthermore, AsymRnR offers significant advantages in processing longer sequences, whereas it provides minor acceleration for image-based DiTs due to their inherently shorter sequence lengths.

## 6. Conclusion

This paper presents AsymRnR, a training-free sampling acceleration approach for video DiTs. AsymRnR decouples sequence length reduction between attention features and allows the reduction scheduling to adaptively distribute reduced computations across blocks and denoising timesteps. To further enhance efficiency, we introduce a matching cache mechanism that minimizes matching overhead, ensuring that acceleration gains are fully realized. Applied to state-of-the-art video DiTs in comprehensive experiments, our approach achieves significant speedups while maintaining high-quality generation. The successful integration with diverse models, including step-distilled models, highlights the generalizability of our approach. These results highlight the potential of AsymRnR to drive practical efficiency improvements in video DiTs generation.

## Acknowledgements

This project is supported by the National Research Foundation, Singapore, under its NRF Professorship Award No. NRF-P2024-001.

## Impact Statement

Generative methods carry the risk of producing biased, privacy-violating, or harmful content. Our method, designed to improve the generation efficiency of video generative models, may also inherit these potential negative impacts. Researchers, users, and service providers should take responsibility for the generated content and strive to ensure positive social impacts.

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

## A. Details of Corollary 3.1

The derivation of the original $k$-NN estimator is detailed in (Wang et al., 2009). For completeness, we present its core concept here. We refer the reader to the cited work for a comprehensive derivation and coverage analysis.

Suppose $p$ and $q$ are the densities of two continuous distributions on $\mathbb{R}^d$, where $p(x) = 0$ almost everywhere $q(x) = 0$. The KL-divergence is defined as:

$$D_{\text{KL}}(p\|q) = \int_{\mathbb{R}^d} p(x) \log \frac{p(x)}{q(x)} \, dx . \quad (6)$$

Let $\{X_i\}_{i=1}^n$ and $\{Y_i\}_{i=1}^m$ be i.i.d. samples drawn from $p$ and $q$, respectively, and $\hat{p}$ and $\hat{q}$ denote consistent estimators of $p$ and $q$. By the law of large numbers, the following statistic provides a consistent estimator for $D_{\text{KL}}(p\|q)$:

$$\frac{1}{n} \sum_{i=1}^n \log \frac{\hat{p}(X_i)}{\hat{q}(X_i)}. \quad (7)$$

To define $\hat{p}$ in Equation (7), consider the closure of the $d$-dimensional Euclidean ball $B(X_i, \rho_k(i))$, centered at $X_i$ with radius $\rho_k(i)$, where $\rho_k(i)$ is the Euclidean distance to the $k$-th nearest-neighbor of $X_i$ in $\{X_j\}_{j\neq i}$. Since $p$ is a continuous density, the ball $B(X_i, \rho_k(i))$ contains $k$ samples from $\{X_j\}_{j\neq i}$ almost surely. The density estimate of $p$ at $X_i$ is

$$\hat{p}(X_i) = \frac{k}{n-1} \frac{1}{c_1(d)\rho_k^d(i)}, \quad (8)$$

where $c_1(d) = \pi^{d/2}/\Gamma(d/2+1)$ is the volume of the unit ball. Similarly the $k$-NN density estimate of $q$ at $X_i$ is

$$\hat{q}(X_i) = \frac{k}{m} \frac{1}{c_1(d)\nu_k^d(i)}, \quad (9)$$

where $\nu_k(i)$ is the Euclidean distance to the $k$-th nearest-neighbor of $X_i$ in $\{Y_j\}_{j=1}^m$. Substituting Equations (8) and (9) into Equation (7) yields the following estimator:

$$\hat{D}_{n,m}(p\|q) = \frac{d}{n} \sum_{i=1}^n \log \frac{\nu_k(i)}{\rho_k(i)} + \log \frac{m}{n-1}. \quad (10)$$

The original paper (Wang et al., 2009) demonstrated that, for any fixed $k$, the bias and variance of the estimator diminish as the sample size $n$ increases. Therefore, $k$ is omitted in Corollary 3.1 for simplicity.

The i.i.d. assumption in the derivation process, together with the convergence analysis from Theorems 1 and 2 in (Wang et al., 2009), is used to apply the law of large numbers. This assumption can be relaxed to the weaker condition of covariance stationarity by applying Chebyshev's law of large numbers, which leads to the same conclusion and facilitates the derivation of Corollary 3.1.

## B. Hyperparameter Tuning

The hyperparameters are manually tuned through only a simple and efficient process, typically within 10 iterations, with each iteration requiring only 1 inference. In practice:

1. We start the first iteration with a low similarity threshold of 0.5 and a low reduction rate of 0.3.

2. We run 1 inference with an arbitrary text prompt. If the generation maintains good, we increase the reduction rate by 0.2 to encourage more aggressive reduction.

3. When a poor generation occurs, we revert to the previous reduction rate, lift the threshold by 0.1, and repeat step 2.

This simple heuristic guides the tuning process with minimal effort. The hyperparameter we used in the experiments will be provided in Appendix C later.

## C. Implementation Details for Experiments

**Video Specification.** In terms of output specifications, CogVideoX-2B and CogVideoX-5B generate 49 frames at a resolution of $480 \times 720$, following their default setting. Mochi-1 generates 85 frames with a resolution of $480 \times 848$. HunyuanVideo produces 129 frames at $544 \times 960$. FastVideo-Hunyuan outputs 65 frames at $720 \times 1280$ resolution. Memory-efficient attention (Rabe & Staats, 2021) is enabled by default in all experiments.

**Diffusion Specification.** The sampling of CogVideoX-2B (Yang et al., 2024) utilizes the 50-step DDIM solver (Song et al., 2021), while CogVideoX-5B employs the 50-step DPM solver (Lu et al., 2022). Mochi-1 (Team, 2024b), HunyuanVideo (Team, 2024c), and FastVideo-Hunyuan (Team, 2024a) use the flow-matching (Esser et al., 2024) Euler solver with 30, 30, and 6 sampling steps, respectively.

**BSM Specification.** To adapt ToMe to the video scenario, we employ a 3D partition with a $2 \times 2 \times 2$ stride in the CogVideoX-2B experiments. This setup ensures consistency with the configuration outlined in Section 3.2. In the experiments involving AsymRnR on Mochi-1, HunyuanVideo, and FastVideo-Hunyuan, the partition stride is expanded to $6 \times 2 \times 2$ to accommodate the higher number of generated frames in these models. As outlined in Section 4.2, we set the matching cache steps to $s = 5$ for CogVideoX variants and $s = 3$ for Mochi-1 and HunyuanVideo. For FastVideo, the matching cache is disabled.

**Similarity Standardization.** The negative Euclidean distance used as the similarity metric spans the range $[0, \infty)$. To improve visualization and usability, as illustrated in Figure 4, we apply a standardization approach. Specifically,

Table 8. **Reduction scheduling details for Section 4.**

| METHOD | FEATURE | REDUCTION |
|---|---|---|
| COGVIDEOX | | |
| + TOME | $H$ | $\{0.0:0.1\}$ |
| + TOME-FAST | $H$ | $\{0.0:0.13\}$ |
| + OURS | $Q$ | $\{0.6:0.4,\ 0.7:0.8\}$ |
| | $V$ | $\{0.8:0.3\}$ |
| + OURS-FAST | $Q$ | $\{0.6:0.6,\ 0.7:0.8\}$ |
| | $V$ | $\{0.7:0.3,\ 0.9:0.4\}$ |
| MOCHI-1 | | |
| + OURS | $Q$ | $\{0.5:0.7\}$ |
| | $V$ | $\{0.7:0.3\}$ |
| + OURS-FAST | $Q$ | $\{0.45:0.6,\ 0.5:0.7\}$ |
| | $V$ | $\{0.7:0.3\}$ |
| HUNYUANVIDEO | | |
| + OURS | $Q$ | $\{0.7:0.9\}$ |
| | $V$ | $\{0.8:0.3\}$ |
| + OURS-FAST | $Q$ | $\{0.5:0.3,\ 0.7:0.9\}$ |
| | $V$ | $\{0.8:0.3\}$ |
| FASTVIDEO | | |
| + OURS | $Q$ | $\{0.7:0.9\}$ |
| | $V$ | $\{0.8:0.3\}$ |
| + OURS-FAST | $Q$ | $\{0.7:0.9\}$ |
| | $V$ | $\{0.8:0.4\}$ |

Table 9. **Destination partition stride and ratio.** Increasing the number of destination tokens excessively leads to higher latency, while reducing them too much compromises video quality.

| STRIDE $(t, h, w)$ | DESTINATION $r_d$ (%) | FLOPs $(\times 10^{15})$ | LATENCY (SECOND) | VBENCH ↑ |
|---|---|---|---|---|
| $(1,2,2)$ | 24.44% | 9.979 | 123.60 | 0.7624 |
| $(2,2,2)$ | 11.28% | 9.917 | 118.16 | **0.7796** |
| $(3,2,2)$ | 7.52% | 9.894 | 115.66 | 0.7417 |
| $(4,2,2)$ | 5.64% | 9.882 | 114.55 | 0.7356 |
| $(2,3,3)$ | 5.13% | 9.879 | 114.10 | 0.7372 |
| $(2,4,4)$ | 2.63% | **9.862** | **112.58** | 0.7231 |

Table 10. **Comparison of additional reduction rates and features.** Simply scaling the reduction rate outperforms the inclusion of $H$ and SymRnR for lower latency.

| FEATURE | FLOPs $(\times 10^{15})$ | LATENCY (SECOND) | VBENCH ↑ |
|---|---|---|---|
| - | 12.000 | 137.30 | 0.8008 |
| $Q + V$ | 9.9755 | 117.29 | **0.7849** |
| $H + Q + V$ | 10.0404 | 117.23 | 0.7766 |

during the registration of similarity distributions in Section 3.4, values outside the 5th–95th percentile range are truncated, followed by min-max standardization. This ensures that all similarity values are scaled to the range $[0, 1]$.

**Reduction Scheduling.** The reduction schedule is defined by two hyperparameters: the similarity threshold for reduction and the reduction rates. The similarity threshold is tuned individually for each DiT model to maintain the quality. Specifically, it is determined through visual inspection of several cases, providing generally effective results, though not necessarily optimal for every sample. On the other hand, the reduction rates are adjusted to achieve the desired acceleration (*e.g.*, a $1.30\times$ speedup).

The reduction schedule is represented as a JSON dictionary. For instance, in CogVideoX-2B AsymRnR, the schedule is specified as $\{'Q': \{0.6:0.4,\ 0.7:0.8\},\ 'V': \{0.8:0.3\}\}$. This indicates that the query sequence will be reduced by 40% when the estimated similarity $\hat{\mathcal{S}}(Q, t, b)$ exceeds $\tau_{Q1} = 0.6$ at timestep $t$ and block $b$, as outlined in Section 3.4. The reduction rate increases to 80% if $\hat{\mathcal{S}}(Q, t, b)$ exceeds $\tau_{Q2} = 0.8$. Similarly, the value (and key) sequences are reduced by 30% when the similarity exceeds $\tau_V = 0.8$. All schedule specifications are summarized in Table 8.

We observed that query sequences are less sensitive to high reduction rates in high-similarity blocks and timesteps, whereas key and value sequences benefit from a more balanced reduction rate across components. This underscores

the importance of our asymmetric design, which enables greater flexibility and optimizes acceleration potential.

## D. More Ablation Results

**Destination Partition.** The partitioning of source and destination tokens in BSM (as detailed in Section 3.1) is critical in final quality. BSM exhibits a complexity of $O(r_d(1 - r_d)n^2)$, which increases monotonically for $0 < r_d < 1/2$, where $r_d$ denotes the fraction of destination tokens and $n$ is the total number of tokens.

As shown in Table 9, a smaller $r_d$ results in significant quality degradation due to less accurate matching, while a larger $r_d$ slightly reduces quality and increases latency due to weakened temporal regularization. We adopt a stride of $(2, 2, 2)$ as the default setting to balance these trade-offs.

**Combining SymRnR and AsymRnR.** A key question is whether increasing the reduction rate or further reducing additional features is more effective in squeezing the latency. Intuitively, SymRnR and AsymRnR can be combined for greater speedup. We explore the parallel integration of SymRnR and AsymRnR: when a block is deemed redundant in $Q$ or $V$ at a given timestep, AsymRnR takes precedence for reduction. Then, the unreduced components can be further processed using SymRnR. The result of this integration is shown in the last row of Table 10. Incorporating additional reductions through SymRnR results in lower quality at the same latency, whereas directly scaling the reduction rate of AsymRnR yields superior performance. We focus on

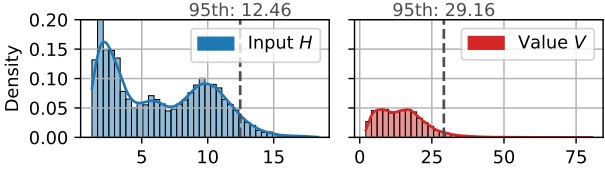

*Figure 10.* **The distribution of feature Euclidean norms.** The dashed line indicates the 95th percentile. Compared to input $H$, the value $V$ norm distribution exhibits a longer tail, which can cause distortion when using cosine similarity for matching.

AsymRnR and leave their integration to future work.

**Similarity Metric for Matching.** The attention operation relies on the dot product metric (*i.e.* cosine similarity) for calculating the attention map (Kim et al., 2017; Dong et al., 2021). Therefore, it is regarded as the standard approach for token similarity in previous works (Bolya et al., 2023; Bolya & Hoffman, 2023; Li et al., 2024; Kahatapitiya et al., 2024a). However, cosine similarity cannot be analyzed through Corollary 3.1 because it lacks metric properties. Empirically, this limitation results in dark spots and a dim appearance in the generated videos, particularly when $V$ is reduced, as quantitatively shown in Table 7.

To investigate the root cause of this issue, we visualized the distribution of feature norms in Figure 10. The norm distributions for $V$ exhibit significantly long tails: a small proportion of tokens show significantly larger norms. Using cosine similarity disregards the magnitude of these tokens, which can result in matching tokens with greatly different magnitudes, thereby causing instability. We use the (negative) Euclidean distance, which effectively captures both directional and magnitude differences between paired vectors, and is supported by a theoretical foundation.

## E. More Results

### E.1. More Qualitative Results

Figure 12 presents additional comparisons of ToMe (Bolya & Hoffman, 2023) on CogVideoX-2B (Yang et al., 2024). Since ToMe is incompatible with CogVideoX-5B, Mochi-1 (Team, 2024b), HunyuanVideo (Team, 2024c), and FastVideo-Hunyuan (Team, 2024a), as discussed in Section 3.3, Figures 11, 13 and 14 illustrate further comparisons of our AsymRnR against the baseline models. More video examples are provided in the Supplementary Materials.

### E.2. More Quantive Results

Table 11 provides additional dimensional metrics of our methods compared to baseline models (Yang et al., 2024; Team, 2024b;c;a) and ToMe (Bolya et al., 2023) on VBench

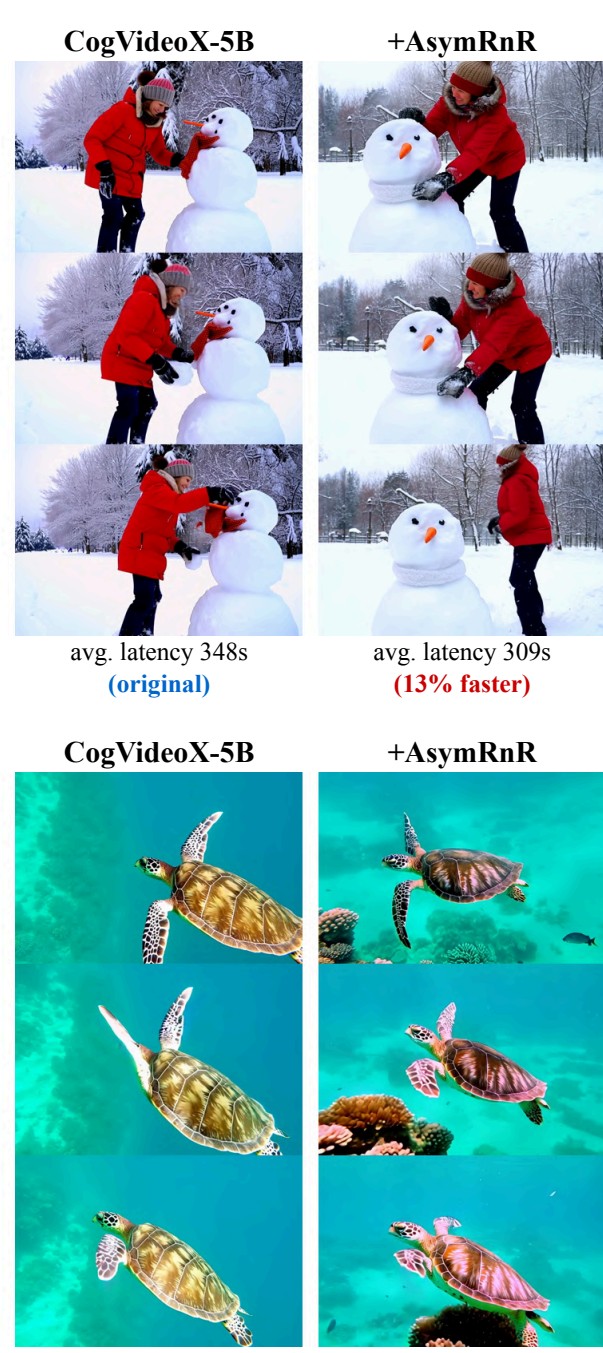

*Figure 11.* **Additional qualitative comparison on CogVideoX-5B** (Yang et al., 2024).

(Huang et al., 2024), serving as an extended reference to Tables 1 and 2.

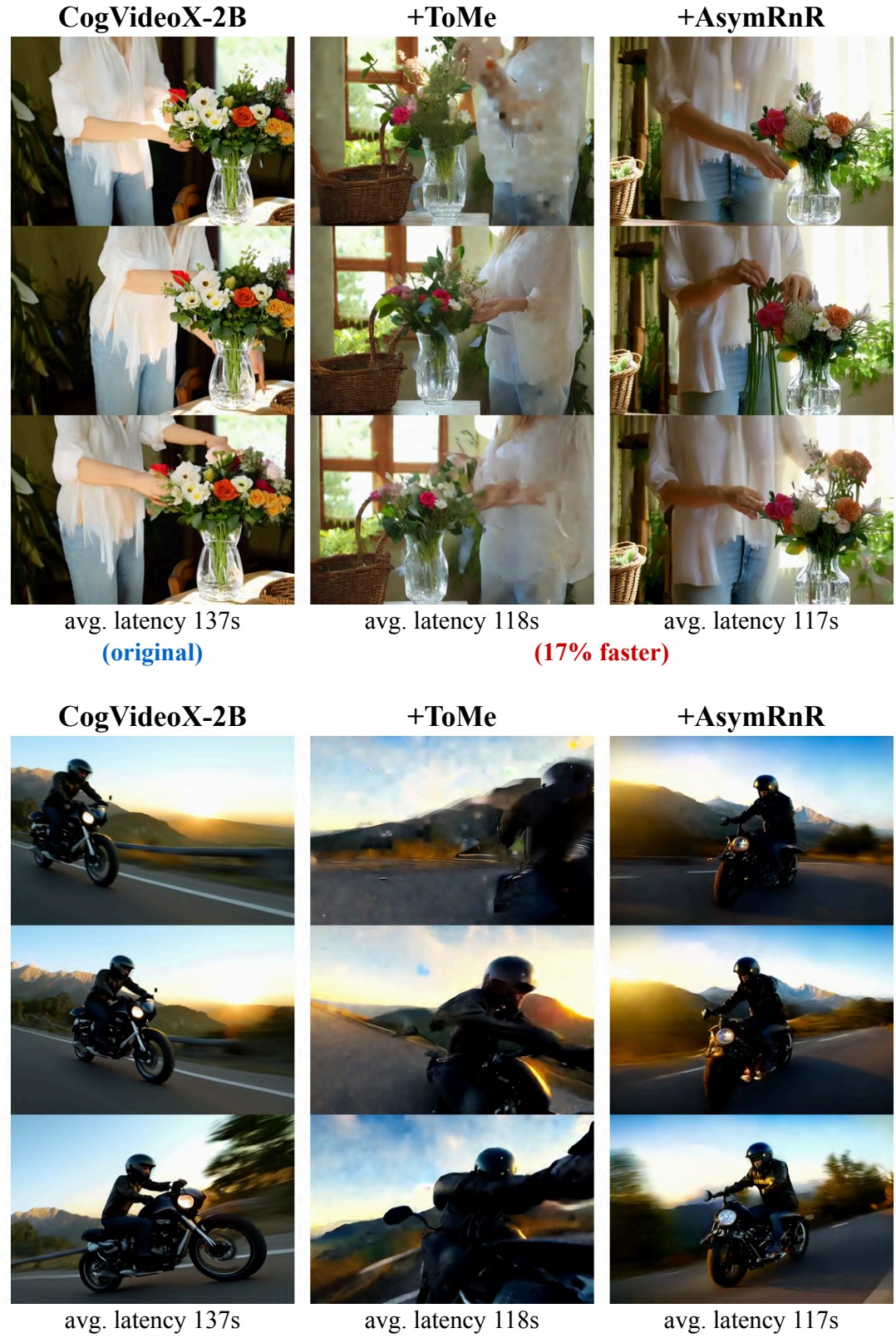

*Figure 12.* **Additional qualitative comparison on CogVideoX-2B (Yang et al., 2024).**

**HunyuanVideo**          **+AsymRnR**

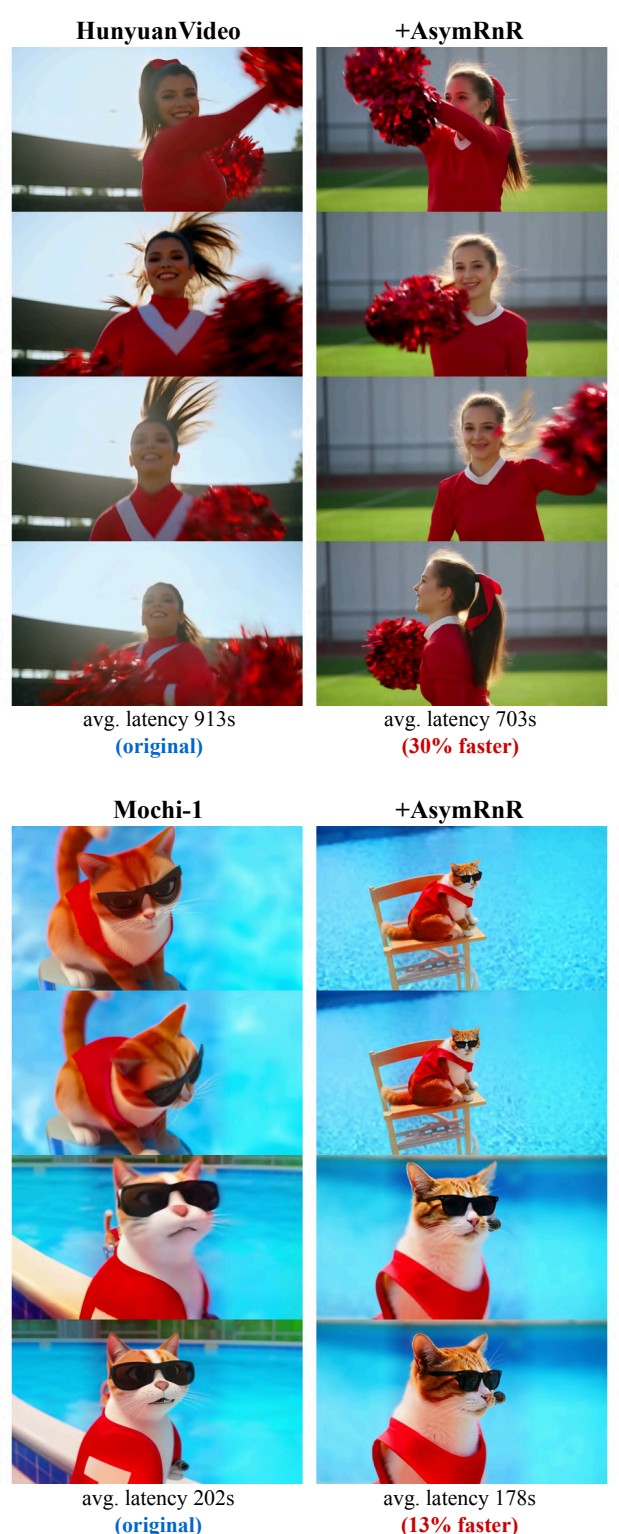

avg. latency 913s
**(original)**

avg. latency 703s
**(30% faster)**

**Mochi-1**          **+AsymRnR**

avg. latency 202s
**(original)**

avg. latency 178s
**(13% faster)**

*Figure 13.* **Additional qualitative comparison on Hunyuan-Video (Team, 2024c) and Mochi-1 (Team, 2024b).**

**FastVideo-Hunyuan**          **+AsymRnR**

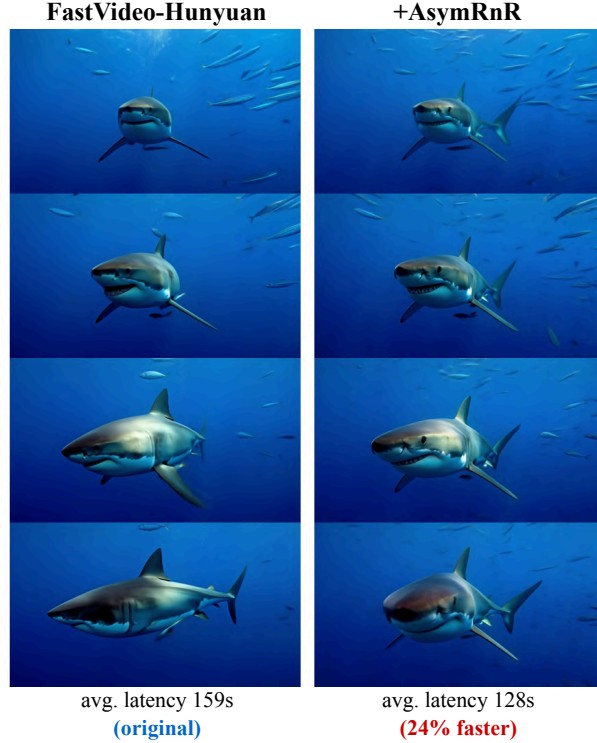

avg. latency 159s
**(original)**

avg. latency 128s
**(24% faster)**

*Figure 14.* **Additional qualitative comparison on FastVideo-Hunyuan (Team, 2024a)**

*Table 11.* **Quantitative results for VBench dimensions (Huang et al., 2024) comparing CogVideoX-2B, CogVideoX-5B (Yang et al., 2024), Mochi-1 (Team, 2024b), HunyuanVideo (Team, 2024c), and FastVideo-Hunyuan (Team, 2024a).**

| METHOD | AESTHETIC QUALITY ↑ | APPEARANCE STYLE ↑ | BACKGROUND CONSISTENCY ↑ | COLOR ↑ | DYNAMIC DEGREE ↑ | HUMAN ACTION ↑ | IMAGING QUALITY ↑ | MOTION SMOOTHNESS ↑ |
|---|---|---|---|---|---|---|---|---|
| COGVIDEOX-2B | 0.6334 | 0.2514 | 0.9390 | 0.8776 | 0.6944 | 0.9400 | 0.6147 | 0.9721 |
| + TOME | 0.6036 | 0.2505 | 0.9328 | 0.8315 | 0.5833 | 0.9600 | 0.5977 | 0.9750 |
| + TOME-FAST | 0.5978 | 0.2500 | 0.9315 | 0.8530 | 0.5000 | 0.9700 | 0.5788 | 0.9759 |
| + OURS | 0.6121 | 0.2504 | 0.9369 | 0.8252 | 0.6667 | 0.9600 | 0.6082 | 0.9699 |
| + OURS-FAST | 0.6099 | 0.2500 | 0.9366 | 0.8762 | 0.5972 | 0.9600 | 0.5986 | 0.9694 |

| METHOD | MUTLIPLE OBJECTS ↑ | OBJECTS CLASS ↑ | OVERALL CONSISTENCY ↑ | SCENE ↑ | SPATIAL RELATIONSHIP ↑ | SUBJECT CONSISTENCY ↑ | TEMPORAL FLICKERING ↑ | TEMPORAL STYLE ↑ |
|---|---|---|---|---|---|---|---|---|
| COGVIDEOX-2B | 0.6502 | 0.8505 | 0.2697 | 0.5378 | 0.6587 | 0.9168 | 0.9711 | 0.2569 |
| + TOME | 0.5473 | 0.8323 | 0.2695 | 0.4586 | 0.7085 | 0.9025 | 0.9747 | 0.2473 |
| + TOME-FAST | 0.5198 | 0.7880 | 0.2665 | 0.4804 | 0.7027 | 0.8949 | 0.9766 | 0.2453 |
| + OURS | 0.5480 | 0.8402 | 0.2750 | 0.5029 | 0.7071 | 0.9184 | 0.9725 | 0.2475 |
| + OURS-FAST | 0.5686 | 0.8188 | 0.2693 | 0.5254 | 0.6957 | 0.9058 | 0.9729 | 0.2443 |

| METHOD | AESTHETIC QUALITY ↑ | APPEARANCE STYLE ↑ | BACKGROUND CONSISTENCY ↑ | COLOR ↑ | DYNAMIC DEGREE ↑ | HUMAN ACTION ↑ | IMAGING QUALITY ↑ | MOTION SMOOTHNESS ↑ |
|---|---|---|---|---|---|---|---|---|
| COGVIDEOX-5B | 0.6350 | 0.2516 | 0.9577 | 0.8002 | 0.6667 | 0.9600 | 0.6342 | 0.9741 |
| + OURS | 0.6298 | 0.2540 | 0.9581 | 0.8300 | 0.6806 | 0.9700 | 0.6284 | 0.9699 |
| + OURS-FAST | 0.6392 | 0.2509 | 0.9588 | 0.8633 | 0.6250 | 0.9700 | 0.6228 | 0.9741 |
| MOCHI-1 | 0.5773 | 0.7989 | 0.9344 | 0.7658 | 0.4375 | 0.9800 | 0.4830 | 0.9537 |
| + OURS | 0.5864 | 0.8008 | 0.9422 | 0.7789 | 0.4028 | 0.9800 | 0.4945 | 0.9712 |
| + OURS-FAST | 0.5870 | 0.8051 | 0.9428 | 0.7505 | 0.4097 | 1.0000 | 0.4906 | 0.9721 |
| HUNYUANVIDEO | 0.6315 | 0.7210 | 0.9392 | 0.8409 | 0.4097 | 0.9700 | 0.5804 | 0.9717 |
| + OURS | 0.6407 | 0.7216 | 0.9473 | 0.8251 | 0.4097 | 0.9700 | 0.6054 | 0.9716 |
| + OURS-FAST | 0.6406 | 0.7216 | 0.9472 | 0.8239 | 0.4097 | 0.9700 | 0.6053 | 0.9716 |
| FASTVIDEO | 0.6318 | 0.7444 | 0.9616 | 0.8857 | 0.4167 | 0.9700 | 0.5783 | 0.9671 |
| + OURS | 0.6270 | 0.7427 | 0.9647 | 0.8819 | 0.4028 | 0.9600 | 0.5662 | 0.9566 |
| + OURS-FAST | 0.6248 | 0.7442 | 0.9617 | 0.8451 | 0.4097 | 0.9600 | 0.5567 | 0.9476 |

| METHOD | MUTLIPLE OBJECTS ↑ | OBJECTS CLASS ↑ | OVERALL CONSISTENCY ↑ | SCENE ↑ | SPATIAL RELATIONSHIP ↑ | SUBJECT CONSISTENCY ↑ | TEMPORAL FLICKERING ↑ | TEMPORAL STYLE ↑ |
|---|---|---|---|---|---|---|---|---|
| COGVIDEOX-5B | 0.6555 | 0.8426 | 0.2694 | 0.5560 | 0.6782 | 0.9227 | 0.9752 | 0.2555 |
| + OURS | 0.6707 | 0.9922 | 0.2717 | 0.5349 | 0.6633 | 0.8981 | 0.9770 | 0.2545 |
| + OURS-FAST | 0.6601 | 0.8180 | 0.2764 | 0.4964 | 0.6448 | 0.9331 | 0.9775 | 0.2564 |
| MOCHI-1 | 0.4794 | 0.8212 | 0.7357 | 0.6417 | 0.6740 | 0.8810 | 0.9683 | 0.7099 |
| + OURS | 0.5465 | 0.8505 | 0.7266 | 0.6408 | 0.6449 | 0.9077 | 0.9705 | 0.6942 |
| + OURS-FAST | 0.5663 | 0.8449 | 0.7308 | 0.6559 | 0.6901 | 0.9068 | 0.9704 | 0.6962 |
| HUNYUANVIDEO | 0.5838 | 0.8560 | 0.7385 | 0.6019 | 0.7253 | 0.9020 | 0.9767 | 0.7013 |
| + OURS | 0.5991 | 0.8655 | 0.7269 | 0.6285 | 0.7662 | 0.9129 | 0.9797 | 0.6977 |
| + OURS-FAST | 0.5983 | 0.8584 | 0.7269 | 0.6196 | 0.7704 | 0.9129 | 0.9797 | 0.6976 |
| FASTVIDEO | 0.6151 | 0.8536 | 0.7308 | 0.6240 | 0.7266 | 0.9213 | 0.9688 | 0.7016 |
| + OURS | 0.5877 | 0.8489 | 0.7378 | 0.6391 | 0.7253 | 0.9206 | 0.9692 | 0.7057 |
| + OURS-FAST | 0.6425 | 0.8457 | 0.7351 | 0.6134 | 0.7208 | 0.9155 | 0.9673 | 0.7055 |

