# OpenReview forum: "AsymRnR: Video Diffusion Transformers Acceleration with Asymmetric Reduction and Restoration"
_ICML.cc/2025/Conference — ICML 2025 poster_

### Official Review · Reviewer_fiPH · 2025-03-09

**Overall Recommendation:** 3

**Summary:**

The authors claim that existing methods for accelerating video DiT sampling often rely on expensive fine-tuning or exhibit limited generalization capabilities. To this end, the authors propose a training-free and model-agnostic method to accelerate video DiTs. Specifically, the authors decouples sequence length reduction between attention features and allows the reduction scheduling to adaptively distribute reduced computations across blocks and denoising timesteps. In addition, the authors introduce a matching cache mechanism to minimize matching overhead. The authors provide extensive experimental validation.

## update after rebuttal
The authors have addressed the majority of my concerns. I therefore maintain my positive rating.

**Claims And Evidence:**

The author's claims are clear and provide sufficient theoretical support.

**Essential References Not Discussed:**

None

**Experimental Designs Or Analyses:**

I have carefully checked the soundness/validity of any experimental designs and analyses, and there are the following problems:

(1) The authors mentioned a variety of Prior Reduction Methods in the paper (such as Bolya & Hoffman, 2023; Li et al., 2024; Kahatapitiya et al., 2024a), but only compared ToMe in the experimental analysis. More experimental comparisons are needed to verify the effectiveness of the paper's methods.

(2) In Fig. 2, the author conducted a comparative experiment on Q in shallow/medium/deep and early/late. I want to know how K and V perform in these positions.

**Methods And Evaluation Criteria:**

The proposed method and evaluation datasets used by the authors are reasonable.

**Other Comments Or Suggestions:**

None

**Other Strengths And Weaknesses:**

None

**Questions For Authors:**

(1) The authors mentioned a variety of Prior Reduction Methods in the paper (such as Bolya & Hoffman, 2023; Li et al., 2024; Kahatapitiya et al., 2024a), but only compared ToMe in the experimental analysis. More experimental comparisons are needed to verify the effectiveness of the paper's methods.

(2) In Fig. 2, the author conducted a comparative experiment on Q in shallow/medium/deep and early/late. I want to know how K and V perform in these positions.

**Relation To Broader Scientific Literature:**

The paper provides new improvement ideas for existing methods and provides new potential for improving the actual generation efficiency of video DiTs.

**Theoretical Claims:**

I have carefully checked the correctness of the proofs for theoretical claims and found no relevant problems.

---

> ### Author Rebuttal · Authors · 2025-03-31
>
> We sincerely thank the Reviewer **fiPH** for the valuable questions and comments. For the concerns and questions, here are our responses, along with supplementary figures and tables available at https://anon0728.github.io/icml-230-supplementary:
>
> **Q1**: The authors mentioned a variety of Prior Reduction Methods (such as Bolya & Hoffman, 2023 [1]; Li et al., 2024 [6]; Kahatapitiya et al., 2024a [7]) in the paper, but only compared ToMe [1] in the experimental analysis. More experimental comparisons are needed to verify the effectiveness of the paper's methods.
>
> **A1**: We thanks for the comment. While various prior reduction methods have been proposed, they are not directly designed for text-to-video diffusion models. Some approaches [1, 2, 3] target discriminative tasks such as image classification, while others [4, 5] focus on autoregressive generation in NLP. These methods typically truncate sequence lengths, making them incompatible with diffusion denoising tasks, which require the input and output sequence lengths to remain identical, as discussed in Sec 2.3.
>
> Li et al. [6] and Kahatapitiya et al. [7], on the other hand, focus on adapting image diffusion models for video editing tasks. However, their task setups and model designs differ significantly from native video diffusion models, making direct comparison inappropriate.
>
> Sequence length reduction for accelerating video diffusion remains relatively underexplored, despite its high potential as demonstrated in our work. Therefore, we primarily compare against ToMe, the only applicable baseline—even though it was originally proposed for image diffusion. Additionally, we evaluate and integrate our method with other acceleration techniques, such as the step-distilled FastVideo method (in Sec 4). We further demonstrate compatibility with caching-based methods through additional experiments in the **A3** of our response to Reviewer **wNPy**, which show that AsymRnR can provide further acceleration when combined with such techniques.
>
> **Reference**
>
> [1] Bolya, Daniel, and Judy Hoffman. "Token merging for fast stable diffusion." CVPR, 2023.
>
> [2] Koner, Rajat, et al. "Lookupvit: Compressing visual information to a limited number of tokens." ECCV, 2024.
>
> [3] Rao, Yongming, et al. "Dynamicvit: Efficient vision transformers with dynamic token sparsification." NIPS, 2021.
>
> [4] Leviathan, Yaniv, Matan Kalman, and Yossi Matias. "Selective attention improves transformer." preprint, 2024.
>
> [5] Xiao, Guangxuan, et al. "Duoattention: Efficient long-context llm inference with retrieval and streaming heads." preprint, 2024.
>
> [6] Li, Xirui, et al. "Vidtome: Video token merging for zero-shot video editing." CVPR. 2024.
>
> [7] Kahatapitiya, Kumara, et al. "Object-centric diffusion for efficient video editing." ECCV, 2024.
>
> ---
>
> **Q2**: In Fig. 2, the author conducted a comparative experiment on Q in shallow/medium/deep and early/late. I want to know how K and V perform in these positions.
>
> **A2**: Thank you for the comment. This observation also motivated our curiosity during the analysis, and we conducted a similar analysis on the $K$ and $V$. Since the analysis is based on random perturbation and $K$ and $V$ tokens are one-to-one corresponding, perturbing $K$ or $V$ yields equivalent effects. The feature-type sensitivity of matching-based (non-random) reduction is further analyzed in Sec 4.3.
>
> The performance trends for $K$ and $V$ mirror those of $Q$ mentioned in Sec 1, but the degradation is less obvious:
>
> 1. Perturbations in later blocks result in greater performance degradation.
> 2. Perturbing early timesteps primarily affects semantic accuracy, while perturbing later timesteps degrades visual details.
>
> Qualitative results are provided in the **supplementary Fig 6**. To improve readability and due to page constraints, we present only the main analysis results in the **main manuscript Fig 2**. We will include the full analysis in the revision.

---

> > ### Comment · Reviewer_fiPH · 2025-04-08
> >
> > Thank you to the authors for addressing all of my concerns. I have no further questions and believe this work makes a clear contribution to the community. I therefore maintain my positive rating.

---

> > > ### Author Response · Authors · 2025-04-08
> > >
> > > Dear reviewer fiPH,
> > >
> > > We would like to express our sincere gratitude to you for acknowledging our work and providing constructive suggestions.
> > >
> > > Many thanks for the time and effort you took to review our work.
> > >
> > > The Authors

---

### Official Review · Reviewer_Mipk · 2025-03-14

**Overall Recommendation:** 3

**Summary:**

This paper studies the importance of different components in video DiTs and proposes Asymmetric Reduction and Restoration (AsymRnR) as a plug-and-play approach to accelerate video DiTs based on previous findings. Experiments on multiple open-source video generation models demonstrate the effectiveness of the proposed method.

## update after rebuttal
The additional experiments of u-net based structures and integration with existing speedup methods address most of my concerns. Therefore, the rating is updated.

**Claims And Evidence:**

Yes, the claims made in the submission are supported by clear and convincing evidence.

**Essential References Not Discussed:**

N/A

**Experimental Designs Or Analyses:**

Yes, I have checked the soundness of the experimental designs and analyses. Seems fine to me.

**Methods And Evaluation Criteria:**

Yes, the proposed methods and evaluation criteria make sense for the problem or application at hand.

**Other Comments Or Suggestions:**

N/A

**Other Strengths And Weaknesses:**

Strengths:
1. The paper provides analysis on the importance of different components in video DiT and provides several key insights which may benefit latter works when designing the methods.
2. The authors provide extensive analysis on different open-source video generation models and qualitative / quantitative evaluation validates the effectiveness of the proposed method.
3. The paper is well-organized and the writing is clear.

Weaknesses:
1. While the proposed method has shown some improvement over the baseline method ToMe, ToMe is original proposed on U-Net based architectures and the comparisons are only conducted on DiT based models. It can be explained by that AsymRnR could only work on transformer structure which contains the designs of Q, K, V which has certain limitation in the generalization ability.
2. Although the proposed method can boost the efficiency of current video generation models to some extend, the improvement in efficiency seems limited compared to other lines of methods, such as distillation methods or feature caching techniques, which could achieve beyond 10x acceleration.
3. The proposed method seems to be an improved version of ToMe based on the empirical findings in video DiT models, the technical contribution needs further justification.

**Questions For Authors:**

1. It is noted that the authors choose different models of CogVideoX in Tab. 1 and Tab. 2, is there a specific reason for this experimental setup?
2. Shown in Tab. 2, the proposed method could result in improvement in VBench scores at certain cases and it is suggested to provide some justification why this could happen. One possibility is that it is caused by the variance of VBench scores, and repeated evaluation may help to eliminate this effect.
3. While the author mentioned that 'Latency is measured using an NVIDIA A100 for CogVideoX variants and an NVIDIA H100 for the rest of models', it would be better to provide some justification on this setup.

**Relation To Broader Scientific Literature:**

This work shares similar motivation with ToMe, i.e., reducing the tokens to enable efficient inference, but exhibits some improvement on DiT-based video generation models.

**Theoretical Claims:**

No.

---

> ### Author Rebuttal · Authors · 2025-03-31
>
> We sincerely thank the Reviewer **Mipk** for the valuable questions and comments. For the concerns and questions, here are our responses, along with supplementary figures and tables available at https://anon0728.github.io/icml-230-supplementary:
>
> **Q1**: The authors choosed different models of CogVideoX in Tab 1 and Tab 2. Is there a specific reason for this experimental setup?
>
> **A1**: Thank you for the comment. CogVideoX 2B and 5B are distinct models with different architectures. We include both to comprehensively evaluate the effectiveness of AsymRnR.
>
> As clarified in **P6 (left, line 322-326)**, ToMe is only compatible with CogVideoX 2B and cannot be applied to other models, such as CogVideoX 5B and HunyuanVideo. In contrast, AsymRnR is compatible with all these models. Therefore, the comparison with ToMe is presented in **Tab 1**, while results on the other models are reported in **Tab 2**.
>
> ---
>
> **Q2**: Shown in Tab 2, the proposed method could result in improvement in VBench scores at certain cases and it is suggested to provide some justification why this could happen. One possibility is that it is caused by the variance of VBench scores, and repeated evaluation may help to eliminate this effect.
>
> **A2**: We agree that variance exists in VBench results, as is common in generative benchmarks. However, the experiments on VBench has already includes over 950 text prompts, and for each prompt, 5 videos have already been generated to mitigate the impact of randomness.
>
> Another possible explanation is the inherent redundancies in the overparameterized models, which may introduce minor negative effects. AsymRnR prunes these redundancies, potentially leading to slight improvements. This hypothesis is empirically supported by **Tab 1 and 2**, where AsymRnR exhibits minimal degradation in larger models with higher FLOPs—and in some cases, performance gains. **The supplementary Fig 1** also show cases where AsymRnR improves the baseline generation.
>
> ---
>
> **Q3**: It is mentioned that 'Latency is measured using an NVIDIA A100 for CogVideoX variants and a H100 for the rest of models', it would be better to provide some justification on this setup.
>
> **A3**: Thank you for the question. Both the baseline models and AsymRnR impose no constraints on the underlying hardware. The experiments were conducted on different devices purely due to the availability of hardware at the time.
>
> ---
>
> Additionally, we would like to clarify a few points raised by Reviewer **Mipk**.
>
> ---
>
> **Q4**: AsymRnR could only work on transformer structure. It has certain limitation in the generalization to UNet backboned video diffusion models.
>
> **A4**: AsymRnR is designed to operate on attention layers, which are commonly present in diffusion models—including UNet models such as Stable Diffusion.
>
> Due to space constraints, we refer the readers to our **A4** response to Reviewer **LhNZ**, which includes additional experiments on the UNet-based video diffusion, AnimateDiff.
>
> ---
>
> **Q5**: The improvement in efficiency seems limited compared to other lines of methods, such as distillation methods or feature caching techniques, which could achieve beyond 10x acceleration.
>
> **A5**: To the best of our knowledge, open-sourced step-distilled video DiTs (eg, FastVideo) can achieve approximately 5× speedup but require substantial training resources. Feature caching methods generally yield around 1.3× acceleration in video DiTs, as shown in **A3** response to Reviewer **wNPy**—on par with AsymRnR.
>
> Additionally, AsymRnR is complementary to these methods and can be integrated with them for additional acceleration.
>
> - The integration with the step-distilled FastVideo method is presented in Sec 4, achieving a total **6.18× speedup over HunyuanVideo**.
> - Integration with the caching-based method PAB results in a **1.71× speedup** without visible distortions.
>
> We also refer Reviewer **Mipk** to response A3 to Reviewer **wNPy** for detailed results.
>
> ---
>
> **Q6**: The proposed method seems to be an improved version of ToMe based on the empirical findings in video DiT models.
>
> **A6**: Both ToMe and AsymRnR accelerate through reducing the number of tokens. However, ToMe relies heavily on heuristic designs and lacks theoretical foundation (eg, the use of cosine similarity). In contrast, we provide a theoretical justification for the matching-based reduction methods in **Corollary 3.1**, which directly informs our design choices (eg, the matching metric in Tab 5).
>
> Furthermore, inspired by the QKV-specific behavior from our exploration, we propose several key components: asymmetric strategy (Sec 3.3), scheduling mechanism (Sec 3.4), and the matching cache (Sec 3.5).
>
> Together, our theoretical insights, empirical analysis, and extensive experiments drive the design of AsymRnR and lay the groundwork for future research across a broader range of token reduction methods.

---

### Official Review · Reviewer_LhNZ · 2025-03-14

**Overall Recommendation:** 3

**Summary:**

The paper presents AsymRnR, a method to accelerate video DiTs without requiring retraining. It exploits the variability in redundancy among different feature tokens across various model blocks and denoising steps. By asymmetrically reducing the computational load during the attention operations, AsymRnR achieves speedups with small loss in output quality. It integrates seamlessly with existing state-of-the-art DiT architectures, enhancing their efficiency across multiple benchmarks.

**Claims And Evidence:**

The paper makes several key claims, which are well-supported by theoretical analysis and experimental results:

1. Claim: AsymRnR provides significant speedup in video DiTs without retraining.
Evidence: Experiments on multiple SOTA models show 24–30% reduction in latency while maintaining high perceptual quality.

2. Claim: The asymmetric reduction strategy improves efficiency while minimizing quality loss.
Evidence: Ablation studies demonstrate that reducing Q tokens too aggressively degrades quality, while reducing K&V is more forgiving. The asymmetric approach balances quality and efficiency better than prior uniform reduction methods (e.g., ToMe).

3. Claim: Matching cache reduces computational overhead while maintaining accuracy.
Evidence: Ablations in Table 4 show that increasing cache steps significantly reduces latency (from 134s to 118s) with only minor quality degradation.

**Essential References Not Discussed:**

None.

**Experimental Designs Or Analyses:**

Potential concerns:
1. The paper does not explicitly discuss the worst-case computational overhead introduced by matching cache and dynamic scheduling.
2. The choice of similarity thresholds for reduction scheduling is not well-explained—how were these hyperparameters tuned?

**Methods And Evaluation Criteria:**

1. The benchmark datasets and evaluation metrics are appropriate for assessing video generation quality and efficiency.
2. The method is tested on multiple state-of-the-art DiT architectures, and follows standard evaluation protocols for video generation.

**Other Comments Or Suggestions:**

1. The paper could clarify the trade-offs between speedup and quality in more detail—e.g., when does AsymRnR start degrading output?
2. A qualitative analysis of artifacts introduced by aggressive reduction would be useful.

**Other Strengths And Weaknesses:**

Strengths:
1. No fine-tuning required, making the method easily adaptable.
2. The aymmetric design is reasonsable with theoretical analysis.
3. Experiments conducted on various SOTA models have demonstrated strong performance. The ablation study is well-conducted.
4. The overall writing is well and logical flow is clear.

Weakness:
1. Hyperparameter sensitivity: Similarity thresholds for reduction scheduling lack clear explanation.
2. Computational overhead of matching cache should be discussed more explicitly.
3. Limited discussion on worst-case performance—are there cases where AsymRnR degrades performance?

**Questions For Authors:**

1. How are the similarity thresholds for reduction scheduling chosen? Are they manually tuned per model, or is there an automated selection process?
2. What is the computational overhead of the matching cache? Does it introduce significant latency in some cases?
3. Does AsymRnR ever degrade performance compared to a baseline? If so, under what conditions?
4. Could the method be extended to latent-space DiTs (e.g., Video LDMs)?

**Relation To Broader Scientific Literature:**

The paper builds on prior token reduction techniques (e.g., ToMe (Bolya & Hoffman, 2023)) but extends them to video DiTs with asymmetric scheduling and caching. Connections to diffusion model acceleration (e.g., distillation approaches like InstaFlow (Liu et al., 2024)) are discussed, highlighting how AsymRnR differs by being training-free. The work is related to efficient attention mechanisms (e.g., Linformer, Performer) but is more specialized for diffusion models.

**Theoretical Claims:**

1. The KL divergence-based analysis (Corollary 3.1) is mathematically sound and provides a formal justification for token reduction strategies.
2. The derivation of the Monte Carlo estimator for KL divergence is based on prior work (Wang et al., 2009) and appears correct.
3. The discussion on token similarity metrics (dot product vs. Euclidean distance) is insightful and supported by empirical findings.

---

> ### Author Rebuttal · Authors · 2025-03-31
>
> We sincerely thank the Reviewer **LhNZ** for the valuable questions and comments. For the concerns and questions, here are our responses, along with supplementary figures and tables available at https://anon0728.github.io/icml-230-supplementary:
>
> ---
>
> **Q1**: How were these hyperparameters (similarity thresholds and reduction rate) tuned? Are they manually tuned per model, or is there an automated selection process?
>
> **A1**: Thank you for the comment. The hyperparameters are manually tuned through only a simple and efficient process——typically within 10 iterations, with each iteration requiring only 1 inference. In practive:
>
> 1. We start the first iteration with a low similarity threshold of 0.5 and a low reduction rate of 0.3.
> 2. We run 1 inference with an arbitrary text prompt. If the generation maintains good, we increase the reduction rate by 0.2 to encourage more aggressive reduction.
> 3. When a poor generation occurs, we revert to the previous reduction rate, lift the threshold by 0.1, and repeat the step 2.
>
> Moreover, hyperparameter re-tuning is not always necessary. In practice, we are able to reuse the same hyperparameters across different model architectures and even models employing different diffusion schedulers. Due to space limitations, we kindly refer Reviewer **LhNZ** to our **A3** response to Reviewer **wNPy** for further details.
>
> This simple heuristic guides the tuning process with minimal effort. We will include the hyperparameter tuning process in the revision.
>
> ---
>
> **Q2**: What is the computational overhead of the matching cache? Does it introduce significant latency in some cases?
>
> **A2**: The matching cache itself does not introduce additional computation. The complexity of a single matching step is analyzed in Appendix C and depends solely on the video size, which is typically fixed to the training resolution. It doesn’t depend on text prompt or reduction rate. With a matching cache step of $s$, the total matching cost can be further reduced by a factor of $1/s$.
>
> In practice, the matching overhead takes approximately **7 seconds** for each generation in the HunyuanVideo experiments (see Sec 4 and Appendix B for more detailed configurations), which is negligible compared to the **over 200 seconds of total acceleration** achieved.
>
> ---
>
> **Q3**: Does AsymRnR ever degrade performance compared to a baseline? If so, under what conditions?
>
> **A3**: Yes, AsymRnR may lead to visible quality degradation.
>
> 1. Under aggressive reduction settings, such as extremely low similarity thresholds or high reduction rates, AsymRnR may introduce distortions, pixelation, or blurring.
>     1. The quantitative analysis for varying reduction rates is provided in Sec 4.3.
>     2. In addition, the **supplementary Fig 2** shows a qualitative study, where we vary the similarity threshold and reduction rate of HunyuanVideo. Under aggressive reduction settings (eg, similarity threshold 0.6 and reduction rate 0.7), noticeable distortion is observed.
> 2. Additionally, when the baseline model already produces unsatisfactory outputs (eg, under extremely fast motion or characters, as shown in the **supplementary Fig 3**), AsymRnR may amplify these issues. However, in most cases where the baseline performs well, AsymRnR maintains stable performance without introducing topic-specific degradation.
>
> We will include the bad case analysis in the revision.
>
> ---
>
> **Q4**: Could the method be extended to latent-space DiTs (e.g., Video LDMs)?
>
> **A4**: The experiments in Sec 4 show our applications of AsymRnR on latent DiTs such as CogVideoX and HunyuanVideo. AsymRnR can also be extended to UNet-based video diffusion models, as these models include attention blocks where AsymRnR operates.
>
> As Video LDM is not open-soruced, we use AnimateDiff to present our extensibility. We apply AsymRnR to the spatial self-attention modules in the highest-resolution stages of AnimateDiff. The corresponding quantitative and qualitative results are presented in the **supplementary Fig 5 and Tab 2.** We achieve 1.20x speedup, with invisable quality degradation. We will include the additional UNet-based experiments in the revision.
>
> Note that, UNet models adopt factorized spatiotemporal blocks have worse performance than the full 3D DiTs at the same parameter scale, as shown in the **supplementary Tab 3**. And it has seen less adopted since early 2024. Our work primarily focuses on SOTA video DiTs in the main manuscript. Nonetheless, AsymRnR remains compatible with the legacy UNet-based video diffusion models.

---

### Official Review · Reviewer_wNPy · 2025-03-17

**Overall Recommendation:** 3

**Summary:**

This paper proposes to asymmetrically reduce the sequence length of attention features to accelerate video DiTs. The proposed approach, called AsymRnR, leverages the observation that different components and stages exhibit varying levels of redundancy. The method introduces a reduction schedule to adaptively distribute reductions across components and a matching cache to enhance efficiency. The authors demonstrate the effectiveness on several video DiTs.

**Claims And Evidence:**

Most claims made in the submission are generally well-supported. However, the paper claims that AsymRnR achieves "negligible degradation in output quality" in some cases and even improves it. While the experimental results show high VBench scores and low LPIPS values, some visual cases in Figure.1, 6, and 7 show misaligned motions with the original results. I wonder how to align these results in practice.

**Essential References Not Discussed:**

None

**Experimental Designs Or Analyses:**

Yes. The authors compare AsymRnR with existing token reduction methods and show performance improvements in terms of efficiency and quality. However, the speedups are very limted, at most 1.3x in Tab.

**Methods And Evaluation Criteria:**

The proposed methods and evaluation criteria are appropriate for the problem of accelerating video diffusion transformers.

**Other Comments Or Suggestions:**

Misalignment of the semantics, the limited speedups and the combination of distillation methods are important for my assesment. I suggest the authors to solve these issues during the rebuttal.

**Other Strengths And Weaknesses:**

## Strengths:
* The proposed method of asymmetric reduction is reasonable.
* The paper provides theoretical analysis to support the proposed reduction strategy, enhancing the credibility of the approach.
* The paper is easy to follow.

## Weaknesses:
*  Despite maintaining semantic consistency, the generated videos may exhibit visual discrepancies compared to baseline models.
* The performance of AsymRnR depends on hyperparameter configurations (e.g., similarity thresholds and reduction rates), which may require tuning for different models.
* The acceleration effects shown in Tables 1 and 2 are very limited, **with a maximum speedup of only 1.3 times**. Can the caching method be combined with distillation methods? For example, **can the caching approach be applied on top of a model that has already been accelerated through distillation to achieve further speedup**?

**Questions For Authors:**

1. What is the meaning of (d) in Equation 2?

2. Figures 1, 6, and 7 may alter the semantic content of the video.

3. In Equation 5, (S) needs to be calculated using the formula in Section 3.5. After calculation, how is ($\hat{S}$) determined for \( H \)/\( Q \)/\( K \)/\( V \)? Is it simply the average value?

**Relation To Broader Scientific Literature:**

Diffusion models, efficient neural network architectures, and video generation acceleration.

**Theoretical Claims:**

The paper includes theoretical analysis to motivate the reduction strategy, specifically through the estimation of KL divergence using Monte Carlo methods.

---

> ### Author Rebuttal · Authors · 2025-03-31
>
> We sincerely thank the Reviewer **wNPy** for the valuable questions and comments. For the concerns and questions, here are our responses, along with supplementary figures and tables available at https://anon0728.github.io/icml-230-supplementary:
>
> **Q1**: While maintaining semantic consistency and the experimental results show high VBench scores and low LPIPS values, some visual cases in **Fig 1, 6, 7** show visual discrepancies (eg, misaligned motions) to baseline models. I wonder how to align these results in practice.
>
> **A1**: Thank you for the comment. We agree that there are visual discrepancies; however, no unique ground-truth video exists for a given text prompt, multiple generations can be equally satisfactory. Therefore, visual quality and textual alignment (measured by VBench score) are the primary performance metrics.
>
> Note that some visual discrepancies occur in cases where AsymRnR produces better results than the baseline model, as illustrated in **supplementary Fig 1**, despite we were not intending to do that. A similar phenomenon is also reported in related works (eg, *Selective Attention Improves Transformer*).
>
> We include baseline generations in the figures to demonstrate that AsymRnR achieves comparable visual quality and textual alignment in most cases, without implying that the outputs should be visually identical. We will include the text prompts alongside the figures in the revision to avoid potential misunderstanding.
>
> ---
>
> **Q2**: The performance of AsymRnR depends on hyperparameter configurations, may require tuning for different models.
>
> **A2**: We acknowledge that the hyperparameters (HPs) searching is performed manually; however, it is very efficient—typically within 10 iterations, with each iteration requiring only 1 inference. Due to the shared concern and word limit, we kindly refer you to our response **A1** to Reviewer **LhNZ** for detailed process of HPs searching.
>
> Moreover, the HPs are transferable:
>
> 1. across text prompts
> 2. across models with
>     1. Different architecture: CogVideo-2B HPs are reused for the 5B variant.
>     2. Different ODE schedulers: FastVideo HPs are transferred to HunyuanVideo.
> 3. Moreover, when integrating AsymRnR with cached methods (detailed in the **A3** below), we can also maintain the same HPs as in Sec 4 and Appendix B.
>
> Notably, other acceleration methods also involve tuning efforts: step distillation approaches require substantial training, and caching-based methods still necessitate model-specific HPs tuning. In comparison, AsymRnR’s HPs tuning is lightweight.
>
> ---
>
> **Q3**: Can the caching method be combined with distillation methods? For example, can the caching approach be applied on top of a model that has already been accelerated through distillation to achieve further speedup?
>
> **A3**: Thank you for the question.
>
> - AsymRnR is compatible with step-distilled models such as FastVideo, as discussed in Sec 4. Notably, despite FastVideo being a distilled 6-step video generation model (5x speed up vs the 30-step HunyuanVideo), AsymRnR achieves a 1.24× further speedup, resulting in a total speedup of **6.18x** over the original HunyuanVideo.
> - Moreover, AsymRnR is also compatible with other caching methods. We compared PAB and PAB + AsymRnR on HunyuanVideo. PAB is configured using the official settings provided on the authors’ GitHub. AsymRnR reuses the HPs from Sec 4 and Appendix B without modification. **The supplementary Tab 1 and Fig 4** shows the compatibility of AsymRnR with the PAB cache method, achieving a total **1.71×** speedup without significant performance loss.
> - To the best of our knowledge, caching-based methods such as PAB are not compatible with step-distilled video diffusions. AsymRnR can be integrated with either caching methods or step-distilled models for acceleration, but their joint integration is beyond the scope of this work.
>
> In summary, although AsymRnR alone does not provide huge acceleration, its compatibility and ease of integration allow it to work seamlessly with other acceleration methods, offering additional benefits.
>
> ---
>
> **Q4**: What is the meaning of $d$ in Eq 2?
>
> **A4**: The $d$ denotes the dimensionality of the vector samples $X_i$, consistent with the definitions throughout the paper. We will explicitly clarify this notation in Collary 3.1 in the revision.
>
> ---
>
> **Q5**: After calculating $S$ using the formula in Sec 3.5, how is $\hat{S}$ in Eq 5 determined for $H, Q, K, V$? Is it simply the average value?
>
> **A5**: Thank you for the comment. The notation $S(A, t, b) = \mathrm{BSM}(A, t, b)$ appears in Sec 3.5 (P6, left column, line 297), where $A \in {H, Q, K, V}$ represents the feature type, and $t$ and $b$ denote the timestep and block. This indicates that $S$ (and $\hat{S}$) is computed separately for each of $H$, $Q$, $K$, and $V$. No averaging or aggregation is applied for our method. We will include the definition of $S$ and the explanation above immediately after Eq 5 in the revision.

---

> > ### Comment · Reviewer_wNPy · 2025-04-07
> >
> > Thanks for the efforts. My concerns have been mostly addressed and I will raise the score.

---

> > > ### Author Response · Authors · 2025-04-07
> > >
> > > Dear reviewer wNPy,
> > >
> > > We would like to express our sincere gratitude to you for acknowledging our work and providing constructive suggestions.
> > >
> > > Many thanks for the time and effort you took to review our work.

---

### Decision · Program_Chairs · 2025-05-01

**Decision:**

Accept (poster)

**Comment:**

The final rating for this paper is 4 weak accept.

Before rebuttal, reviewers' main concerns are: 1) hyperparameter need to be changed for different methods; 2) acceleration effects are very limited, with a maximum speedup of only 1.3x; 3) limited technical contribution; 4) more comparison with other methods are needed.

After rebuttal, all reviewers confirmed they had read the authors' response and would update reviews if needed. Reviewers wNPy suggested their concerns had been mostly addressed and raise the score. Reviewer fiPH mentioned authors addressed all their concersn and maintained rating. Reviewer LhNZ and Mipk didn't give further comments and maintain their score.

Consider all these AC gave weak accept recommendation for this paper which aligned with all the reviewers' rating.